# Robust Learning of Fixed-Structure Bayesian Networks in Nearly-Linear Time

**Yu Cheng**
Department of Mathematics (MSCS)
University of Illinois at Chicago
yucheng2@uic.edu

**Honghao Lin**
Institute for Theoretical Computer Science
Shanghai University of Finance and Economics
Guilan3010@gmail.com

## Abstract

We study the problem of learning Bayesian networks where an $\epsilon$-fraction of the samples are adversarially corrupted. We focus on the fully-observable case where the underlying graph structure is known. In this work, we present the first nearly-linear time algorithm for this problem with a dimension-independent error guarantee. Previous robust algorithms with comparable error guarantees are slower by at least a factor of $(d/\epsilon)$, where $d$ is the number of variables in the Bayesian network and $\epsilon$ is the fraction of corrupted samples. Our algorithm and analysis are considerably simpler than those in previous work. We achieve this by establishing a direct connection between robust learning of Bayesian networks and robust mean estimation. As a subroutine in our algorithm, we develop a robust mean estimation algorithm whose runtime is nearly-linear in the number of nonzeros in the input samples, which may be of independent interest.

## 1 Introduction

Probabilistic graphical models (Koller & Friedman, 2009) offer an elegant and succinct way to represent structured high-dimensional distributions. The problem of inference and learning in probabilistic graphical models is an important problem that arises in many disciplines (see Wainwright & Jordan (2008) and the references therein), which has been studied extensively during the past decades (see, e.g., Chow & Liu (1968); Dasgupta (1997); Abbeel et al. (2006); Wainwright et al. (2006); Anandkumar et al. (2012); Santhanam & Wainwright (2012); Loh & Wainwright (2012); Bresler et al. (2013; 2014); Bresler (2015)).

Bayesian networks (Jensen & Nielsen, 2007) are an important family of probabilistic graphical models that represent conditional dependence by a directed graph (see Section 2 for a formal definition). In this paper, we study the problem of learning Bayesian networks where an $\epsilon$-fraction of the samples are adversarially corrupted. We focus on the simplest setting: all variables are binary and observable, and the structure of the Bayesian network is given to the algorithm.

Formally, we work with the following corruption model:

**Definition 1.1** ($\epsilon$-Corrupted Set of Samples). *Given $0 < \epsilon < 1/2$ and a distribution family $\mathcal{P}$ on $\mathbb{R}^d$, the algorithm first specifies the number of samples $N$, and $N$ samples $X_1, X_2, \ldots, X_N$ are drawn from some unknown $P \in \mathcal{P}$. The adversary inspects the samples, the ground-truth distribution $P$, and the algorithm, and then replaces $\epsilon N$ samples with arbitrary points. The set of $N$ points is given to the algorithm as input. We say that a set of samples is $\epsilon$-corrupted if it is generated by this process.*

This is a strong corruption model which generalizes many existing models. In particular, it is stronger than Huber's contamination model (Huber, 1964), because we allow the adversary to add bad samples and remove good samples, and he can do so adaptively.

Our goal is to design robust algorithms for learning Bayesian networks with dimension-independent error. More specifically, given as input an $\epsilon$-corrupted set of samples drawn from some ground-truth Bayesian network $P$ and the graph structure of $P$, we want the algorithm to output a Bayesian network $Q$, such that the total variation distance between $P$ and $Q$ is upper bounded by a function that depends only on $\epsilon$ (the fraction of corruption) but not $d$ (the number of variables in $P$).

In the fully-observable fixed-structure setting, the problem is straightforward when there is no corruption. We know that the empirical estimator (which computes the empirical conditional probabilities) is sample efficient and runs in linear time (Dasgupta, 1997).

It turns out that the problem becomes much more challenging when there is corruption. Even for robust learning of binary product distributions (i.e., a Bayesian network with an empty dependency graph), the first computational efficient algorithms with dimension-independent error was only discovered in (Diakonikolas et al., 2019a). Subsequently, (Cheng et al., 2018) gave the first polynomial-time algorithms for robust learning of fixed-structured Bayesian networks. The main drawback of the algorithm in (Cheng et al., 2018) is that it runs in time $\Omega(Nd^2/\epsilon)$, which is slower by at least a factor of $(d/\epsilon)$ compared to the fastest non-robust estimator.

Motivated by this gap in the running time, in this work we want to resolve the following question:

> *Can we design a robust algorithm for learning Bayesian networks in the fixed-structure fully-observable setting that runs in nearly-linear time?*

## 1.1 OUR RESULTS AND CONTRIBUTIONS

We resolve this question affirmatively by proving Theorem 1.2. We say a Bayesian network is *c-balanced* if all its conditional probabilities are between $c$ and $1 - c$. For the ground-truth Bayesian network $P$, let $m$ be the size of its *conditional probability table* and $\alpha$ be its *minimum parental configuration probability* (see Section 2 for formal definitions).

**Theorem 1.2** (informal statement). *Consider an $\epsilon$-corrupted set of $N = \widetilde{\Omega}(m/\epsilon^2)$ samples drawn from a $d$-dimensional Bayesian network $P$. Suppose $P$ is $c$-balanced and has minimum parental configuration probability $\alpha$, where both $c$ and $\alpha$ are universal constants. We can compute a Bayesian network $Q$ in time $\widetilde{O}(Nd)$ such that $d_{\mathrm{TV}}(P, Q) \leq \epsilon\sqrt{\ln(1/\epsilon)}$.* [1]

For simplicity, we stated our result in the very special case where both $c$ and $\alpha$ are $\Omega(1)$. Our approach works for general values of $\alpha$ and $c$, where our error guarantee degrades gracefully as $\alpha$ and $c$ gets smaller. A formal version of Theorem 1.2 is given as Theorem 4.1 in Section 4.

Our algorithm has optimal error guarantee, sample complexity, and running time (up to logarithmic factors). There is an information-theoretic lower bound of $\Omega(\epsilon)$ on the error guarantee, which holds even for Bayesian networks with only one variable. A sample complexity lower bound of $\Omega(m/\epsilon^2)$ holds even without corruption (see, e.g., (Canonne et al., 2017)).

**Our Contributions.** We establish a novel connection between robust learning of Bayesian networks and robust mean estimation. At a high level, we show that one can essentially reduce the former to the latter. This allows us to take advantage of the recent (and future) advances in robust mean estimation and apply the algorithms almost directly to obtain new algorithms for learning Bayesian networks.

Our algorithm and analysis are considerably simpler than those in previous work. For simplicity, consider learning binary product distributions as an example. Cheng et al. (2018) tried to remove samples to make the empirical covariance matrix closer to a diagonal matrix (since the true covariance matrix is diagonal because each coordinate is independent). They used a "filtering" approach which requires proving specific tail bounds on the samples. In contrast, we show that it suffices to use any robust mean estimation algorithms which minimize the spectral norm of the empirical covariance matrix (regardless of whether it is close to being diagonal or not).

As a subroutine in our approach, we develop the first robust mean estimation algorithm that runs in nearly input-sparsity time (i.e., in time nearly linear in the total number of nonzero entries in the input), which may be of independent interest. The main computation bottleneck of current nearly-linear time robust mean estimation algorithms (Cheng et al., 2019a; Depersin & Lecué, 2019; Dong et al., 2019) is running matrix multiplication weight update with the Johnson-Lindenstrauss lemma, which we show can be done in nearly input-sparsity time.

---

[1] Throughout the paper, we use $\widetilde{O}(f)$ to denote $O(f \operatorname{polylog}(f))$.

## 1.2 RELATED WORK

**Bayesian Networks.** Probabilistic graphical models (Koller & Friedman, 2009) provide an appealing and unifying formalism to succinctly represent structured high-dimensional distributions. The general problem of inference in graphical models is of fundamental importance and arises in many applications across several scientific disciplines (see Wainwright & Jordan (2008) and references therein). The problem of learning graphical models from data (Neapolitan, 2003; Daly et al., 2011) has many variants: (i) the family of graphical models (e.g., directed, undirected), (ii) whether the data is fully or partially observable, and (iii) whether the graph structure is known or not. This learning problem has been studied extensively (see, e.g., Chow & Liu (1968); Dasgupta (1997); Abbeel et al. (2006); Wainwright et al. (2006); Anandkumar et al. (2012); Santhanam & Wainwright (2012); Loh & Wainwright (2012); Bresler et al. (2013; 2014); Bresler (2015)), resulting in a beautiful theory and a collection of algorithms in various settings.

**Robust Statistics.** Learning in the presence of outliers has been studied since the 1960s (Huber, 1964). For the most basic problem of robust mean estimation, it is well-known that the empirical median works in one dimension. However, most natural generalizations of the median to high dimensions (e.g., coordinate-wise median, geometric median) would incur an error of $\Omega(\epsilon\sqrt{d})$, even in the infinite sample regime (see, e.g., Diakonikolas et al. (2019a); Lai et al. (2016)). After decades of work, sample-efficient robust estimators have been discovered (e.g., the Tukey median (Tukey, 1975; Devroye & Györfi, 1985; Chen et al., 2018)). However, the Tukey median is NP-Hard to compute in the worse case (Johnson & Preparata, 1978; Amaldi & Kann, 1995) and many heuristics for approximating it perform poorly as the dimension scales (Clarkson et al., 1993; Chan, 2004; Miller & Sheehy, 2010).

**Computational Efficient Robust Estimators.** Recent work (Diakonikolas et al., 2019a; Lai et al., 2016) gave the first polynomial-time algorithms several high-dimensional unsupervised learning tasks (e.g., mean and covariance estimation) with dimension-independent error guarantees. After the dissemination of (Diakonikolas et al., 2019a; Lai et al., 2016), algorithmic high-dimensional robust statistics has attracted a lot of recent attention and there has been a flurry of research that obtained polynomial-time robust algorithms for a wide range of machine learning and statistical tasks (see, e.g., Balakrishnan et al. (2017); Charikar et al. (2017); Diakonikolas et al. (2017a;b); Steinhardt et al. (2018); Diakonikolas et al. (2018); Hopkins & Li (2018); Kothari et al. (2018); Prasad et al. (2020); Diakonikolas et al. (2019b); Klivans et al. (2018); Diakonikolas et al. (2019c); Liu et al. (2020); Cheng et al. (2020); Zhu et al. (2020)). In particular, the most relevant prior work is (Cheng et al., 2018), which gave the first polynomial-time algorithms for robust learning of fixed-structure Bayesian networks.

**Faster Robust Estimators.** While recent work gave polynomial-time robust algorithms for many tasks, these algorithms are often significantly slower than the fastest non-robust ones (e.g., sample average for mean estimation). Cheng et al. (2019a) gave the first nearly-linear time algorithm for robust mean estimation and initiated the research direction of designing robust estimators that are as efficient as their non-robust counterparts. Since then, there have been several works that develop faster robust algorithms for various learning and statistical tasks, including robust mean estimation for heavy-tailed distributions Dong et al. (2019); Depersin & Lecué (2019), robust covariance estimation Cheng et al. (2019b); Li & Ye (2020), robust linear regression Cherapanamjeri et al. (2020a), and list-decodable mean estimation Cherapanamjeri et al. (2020b); Diakonikolas et al. (2020).

**Organization.** In Section 2, we define our notations and provide some background on robust learning of Bayesian networks and robust mean estimation. In Section 3, we give an overview of our approach and highlight some of our key technical results. In Section 4, we present our algorithm for robust learning of Bayesian networks and prove our main result.

## 2 PRELIMINARIES

**Bayesian Networks.** Fix a $d$-node directed acyclic graph $H$ whose nodes are labelled $[d] = \{1, 2, \ldots, d\}$ in topological order (every edge goes from a node with smaller index to one with larger index). Let Parents($i$) be the parents of node $i$ in $H$. A probability distribution $P$ on

$\{0,1\}^d$ is a *Bayesian network* (or *Bayes net*) with graph $H$ if, for each $i \in [d]$, we have that $\Pr_{X \sim P}[X_i = 1 \mid X_1, \ldots, X_{i-1}]$ depends only on the values $X_j$ where $j \in \mathrm{Parents}(i)$.

**Conditional Probability Table.** Let $P$ be a Bayesian network with graph $H$. Let $\Gamma = \{(i,a) : i \in [d], a \in \{0,1\}^{|\mathrm{Parents}(i)|}\}$ be the set of all possible parental configurations. Let $m = |\Gamma|$. For $(i,a) \in \Gamma$, the *parental configuration* $\Pi_{i,a}$ is defined to be the event that $X(\mathrm{Parents}(i)) = a$. The conditional probability table $p \in [0,1]^m$ of $P$ is given by $p_{i,a} = \Pr_{X \sim P}[X(i) = 1 \mid \Pi_{i,a}]$.

In this paper, we often index $p$ as an $m$-dimensional vector. We use the notation $p_k$ and the associated events $\Pi_k$, where each $k \in [m]$ stands for an $(i,a) \in \Gamma$ lexicographically ordered.

**Notations.** For a vector $v$, let $\|v\|_2$ and $\|v\|_\infty$ be the $\ell_2$ and $\ell_\infty$ norm of $v$ respectively. We write $\sqrt{v}$ and $1/v$ for the entrywise square root and entrywise inverse of a vector $v$ respectively. For two vectors $x$ and $y$, we write $x^\top y$ for their inner product, and $x \circ y$ for their entrywise product.

We use $I$ to denote the identity matrix. For a matrix $M$, let $M_i$ be the $i$-th column of $M$, and let $\|M\|_2$ be the spectral norm of $M$. For a vector $v \in \mathbb{R}^n$, let $\mathrm{diag}(v) \in \mathbb{R}^{n \times n}$ denote a diagonal matrix with $v$ on the diagonal.

Throughout this paper, we use $P$ to denote the ground-truth Bayesian network. We use $d$ for the dimension (i.e., the number of nodes) of $P$, $N$ for the number of samples, $\epsilon$ for the fraction of corrupted samples, and $m = \sum_{i=1}^d 2^{|\mathrm{Parents}(i)|}$ for the size of the conditional probability table of $P$. We use $p \in \mathbb{R}^m$ to denote the (unknown) ground-truth conditional probabilities of $P$, and $q \in \mathbb{R}^m$ for our current guess of $p$.

Let $G^\star$ be the original set of $N$ uncorrupted samples drawn from $P$. After the adversary corrupts an $\epsilon$-fraction of $G^\star$, let $G \subseteq G^\star$ be the remaining set of good samples, and $B$ be the set of bad samples added by the adversary. The set of samples $S = G \cup B$ is given to the algorithm as input. Let $X \in \mathbb{R}^{d \times N}$ denote the sample matrix whose $i$-th column $X_i \in \mathbb{R}^d$ is the $i$-th input sample. Abusing notation, we sometimes also use $X$ as a random variable (e.g., a sample drawn from $P$).

We use $\pi^P \in \mathbb{R}^m$ to denote the parental configuration probabilities of $P$. That is, $\pi_k^P = \Pr_{X \sim P}[X \in \Pi_k]$. For a set $S$ of samples, we use $\pi^S \in \mathbb{R}^m$ to denote the *empirical* parental configuration probabilities over $S$: $\pi_k^S = \Pr_X[X \in \Pi_k]$ where $X$ is uniformly drawn from $S$.

**Balance and Minimum Configuration Probability.** We say a Bayesian network $P$ is $c$-balanced if all conditional probabilities of $P$ are between $c$ and $1-c$. We use $\alpha$ for the minimum probability of parental configuration of $P$: $\alpha = \min_k \pi_k^P$.

In this paper, we assume that the ground-truth Bayesian network is $c$-balanced, and its minimum parental configuration probability $\alpha$ satisfies that $\alpha = \Omega\big((\epsilon\sqrt{\ln(1/\epsilon)})^{2/3}c^{-1/3}\big)$. Without loss of generality, we further assume that both $c$ and $\alpha$ are given to the algorithm.

## 2.1 Total Variation Distance between Bayesian Networks

Let $P$ and $Q$ be two distributions supported on a finite domain $D$. For a set of outcomes $A$, let $P(A) = \Pr_{X \sim P}[X \in A]$. The total variation distance between $P$ and $Q$ is defined as

$$d_{\mathrm{TV}}(P,Q) = \max_{A \subseteq D} |P(A) - Q(A)| .$$

For two balanced Bayesian networks that share the same structure, it is well-known that the closeness in their conditional probabilities implies their closeness in total variation distance. Formally, we use the following lemma from Cheng et al. (2018), which upper bounds the total variation distance between two Bayesian networks in terms of their conditional probabilities.

**Lemma 2.1** (Cheng et al. (2018)). *Let $P$ and $Q$ be two Bayesian networks that share the same structure. Let $p$ and $q$ denote the conditional probability tables of $P$ and $Q$ respectively. We have*

$$(d_{\mathrm{TV}}(P,Q))^2 \le 2 \sum_k \sqrt{\pi_k^P \pi_k^Q} \frac{(p_k - q_k)^2}{(p_k + q_k)(2 - p_k - q_k)} .$$

## 2.2 EXPANDING THE DISTRIBUTION TO MATCH CONDITIONAL PROBABILITY TABLE

Lemma 2.1 states that to learn a known-structure Bayesian network $P$, it is sufficient to learn its conditional probabilities $p$. However, a given coordinate of $X \sim P$ may contain information about multiple conditional probabilities (depending on which parental configuration happens).

To address this issue, we use a similar approach as in Cheng et al. (2018). We expand each sample $X$ into an $m$-dimensional vector $f(X, q)$, such that each coordinate of $f(X, q)$ corresponds to an entry in the conditional probability table. Intuitively, $q \in \mathbb{R}^m$ is our current guess for $p$, and initially we set $q$ to be the empirical conditional probabilities. We use $q$ to fill in the missing entries in $f(X, q)$ for which the parental configurations fail to happen.

**Definition 2.2.** *Let $f(X, q)$ for $\{0, 1\}^d \times \mathbb{R}^m \to \mathbb{R}^m$ be defined as follows:*

$$f(X, q)_{i,a} = \begin{cases} X_i - q_{i,a} & X \in \Pi_{i,a} \\ 0 & \text{otherwise} \end{cases}$$

When $X \sim P$ and $q = p$, the distribution of $f(X, p)$ has many good properties. Using the conditional independence of Bayesian networks, we can compute the first and second moment of $f(X, p)$ and show that $f(X, p)$ has subgaussian tails.

**Lemma 2.3.** *For $X \sim P$ and $f(X, p)$ as defined in Definition 2.2, we have*

*(i) $\mathbb{E}(f(X, p)) = 0$.*     *(ii) $\mathrm{Cov}[f(X, p)] = \mathrm{diag}(\pi^P \circ p \circ (1 - p))$.*

*(iii) For any unit vector $v \in \mathbb{R}^m$, we have $\mathrm{Pr}_{X \sim P}\left[|v^\top f(X, p)| \geq T\right] \leq 2\exp(-T^2/2)$.*

We defer the proof of Lemma 2.3 to Appendix A. A slightly stronger version of Lemma 2.3 was proved in Cheng et al. (2018), which discusses tail bounds for $f(X, q)$. For our analysis, Lemma 2.3 is sufficient.

For general values of $q$, we can similarly compute the mean of $f(X, q)$:

**Lemma 2.4.** *Let $\pi^P$ denote the parental configuration of $P$. For $X \sim P$ and $f(X, q)$ as defined in Definition 2.2, we have $\mathbb{E}[f(X, q)] = \pi^P \circ (p - q)$.*

## 2.3 DETERMINISTIC CONDITIONS ON GOOD SAMPLES

To avoid dealing with the randomness of the good samples, we require the following deterministic conditions to hold for the original set $G^\star$ of $N$ good samples (before the adversary's corruption).

We prove in Appendix A that these three conditions hold simultaneously with probability at least $1 - \tau$ if we draw $N = \Omega(m \log(m/\tau)/\epsilon^2)$ samples from $P$.

The first condition states that we can obtain a good estimation of $p$ from $G^\star$. Let $p^{G^\star}$ denote the empirical conditional probabilities over $G^\star$. We have

$$\left\| \sqrt{\pi^P} \circ (p - p^{G^\star}) \right\|_2 \leq O(\epsilon) . \tag{1}$$

The second condition says that we can estimate the parental configuration probabilities $\pi^P$ from any $(1 - 2\epsilon)$-fraction of $G^\star$. Formally, for any subset $T \subset G^\star$ with $|T| \geq (1 - 2\epsilon)N$, we have

$$\left\| \pi^T - \pi^P \right\|_\infty \leq O(\epsilon) . \tag{2}$$

The third condition is that the empirical mean and covariance of any $(1 - 2\epsilon)$-fraction of $G^\star$ are very close to the true mean and covariance of $f(X, p)$. Formally, for any subset $T \subset G^\star$ with $|T| \geq (1 - 2\epsilon)N$, we require the following to hold for $\delta_1 = \epsilon\sqrt{\ln 1/\epsilon}$ and $\delta_2 = \epsilon \ln(1/\epsilon)$:

$$\left\| \frac{1}{|T|} \sum_{i \in T} f(X_i, p) \right\|_2 \leq O(\delta_1) , \quad \left\| \frac{1}{|T|} \sum_{i \in T} f(X_i, p) f(X_i, p)^\top - \Sigma \right\|_2 \leq O(\delta_2) , \tag{3}$$

where $\Sigma = \mathrm{Cov}[f(X, p)] = \mathrm{diag}(\pi^P \circ p \circ (1 - p))$.

## 2.4 ROBUST MEAN ESTIMATION AND STABILITY CONDITIONS

Robust mean estimation is the problem of learning the mean of a $d$-dimensional distribution from an $\epsilon$-corrupted set of samples. As we will see in later sections, to robustly learn Bayesian networks, we repeatedly use robust mean estimation algorithms as a subroutine.

Recent work (Diakonikolas et al., 2019a; Lai et al., 2016) gave the first polynomial-time algorithms for robust mean estimation with dimension-independent error guarantees. The key observation in Diakonikolas et al. (2019a) is the following: if the empirical mean is inaccurate, then many samples must be far from the true mean in roughly the same direction. Consequently, these samples must alter the variance in this direction more than they distort the mean. Therefore, if the empirical covariance behaves as we expect it to be, then the empirical mean provides a good estimate to the true mean.

Many robust mean estimation algorithms follow the above intuition, and they require the following stability condition to work (Definition 2.5). Roughly speaking, the stability condition states that the mean and covariance of the good samples are close to that of the true distribution, and more importantly, this continues to hold if we remove any $2\epsilon$-fraction of the samples.

**Definition 2.5** (Stability Condition (see, e.g., Diakonikolas & Kane (2019))). *Fix* $0 < \epsilon < \frac{1}{2}$. *Fix a $d$-dimensional distribution $X$ with mean $\mu_X$. We say a set $S$ of samples is $(\epsilon, \beta, \gamma)$-stable with respect to $X$, if for every subset $T \subset S$ with $|T| \geq (1 - 2\epsilon)|S|$, the following conditions hold:*

*(i)* $\left\| \frac{1}{|T|} \sum_{X \in T} (X - \mu_X) \right\|_2 \leq \beta$ , *(ii)* $\left\| \frac{1}{|T|} \sum_{X \in T} (X - \mu_X)(X - \mu_X)^\top - I \right\|_2 \leq \gamma$ .

Subsequent work (Cheng et al., 2019a; Dong et al., 2019; Depersin & Lecué, 2019) improved the runtime of robust mean estimation to nearly-linear time. Formally, we use the following result from Dong et al. (2019). A set $S$ is an $\epsilon$-corrupted version of a set $T$ if $|S| = |T|$ and $|S \setminus T| \leq \epsilon|S|$.

**Lemma 2.6** (Robust Mean Estimation in Nearly-Linear Time (Dong et al., 2019)). *Fix a set of $N$ samples $G^\star$ in $\mathbb{R}^d$. Suppose $G^\star$ is $(\epsilon, \beta, \gamma)$-stable with respect to a $d$-dimensional distribution $X$ with mean $\mu_X \in \mathbb{R}^d$. Let $S$ be an $\epsilon$-corrupted version of $G^\star$. Given as input $S$, $\epsilon$, $\beta$, $\gamma$, there exists an algorithm that can output an estimator $\hat{\mu} \in \mathbb{R}^d$ in time $\widetilde{O}(Nd)$, such that with high probability, $\|\hat{\mu} - \mu_X\|_2 \leq O(\sqrt{\epsilon\gamma} + \beta + \epsilon\sqrt{\log 1/\epsilon})$ .*

As we will see later, a black-box use of Lemma 2.6 does not give the desired runtime in our setting. Instead, we extend Lemma 2.6 to handle sparse input such that it runs in time nearly-linear in the number of non-zeros in the input (see Lemma 3.3).

## 3 OVERVIEW OF OUR APPROACH

In this section, we give an overview of our approach and highlight some of our key technical results.

To robustly learn the ground-truth Bayesian network $P$, it is sufficient to learn its conditional probabilities $p \in \mathbb{R}^m$. At a high level, we start with a guess $q \in \mathbb{R}^m$ for $p$ and then iteratively improve our guess to get closer to $p$. For any $q \in \mathbb{R}^m$, we can expand the input samples into $m$-dimensional vectors $f(X, q)$ as in Definition 2.2. We first show that the expectation of $f(X, q)$ gives us useful information about $(p - q)$.

Recall that $\pi^P$ is the parental configuration probabilities of $P$. By Lemma 2.4, we have

$$\mathbb{E}_{X \sim P}[f(X, q)] = \pi^P \circ (p - q) .$$

Note that if we had access to this expectation and the vector $\pi^P$, we could recover $p$ immediately: we can set $q' = \mathbb{E}[f(X, q)] \circ (1/\pi^P) + q$ which simplifies to $q' = p$.

Note that since $S$ is an $\epsilon$-corrupted set of samples of $P$, we know that $\{f(X_i, q)\}_{i \in S}$ is an $\epsilon$-corrupted set of samples of the distribution $f(X, q)$ (with $X \sim P$). Therefore, we can run robust mean estimation algorithms on $\{f(X_i, q)\}_{i \in S}$ to learn $\mathbb{E}[f(X, q)]$. It turns out a good approximation of $\mathbb{E}[f(X, q)]$ can help us improve our current guess $q$.

There are two main difficulties in getting this approach to work.

The first difficulty is that, to use robust mean estimation algorithms, we need to show that $f(X, q)$ satisfies the stability condition in Definition 2.5. This requires us to analyze the first two moments and tail bounds of $f(X, q)$. Consider the second moment for example. Ideally, we would like to have a statement of the form $\mathrm{Cov}[f(X, q)] \approx \mathrm{Cov}[f(X, p)] + (p - q)(p - q)^\top$, but this is false because we only have $f(X, p)_k - f(X, q)_k = (p - q)_k$ if the $k$-th parental configuration happens for $X$. Intuitively, the "error" $(p - q)$ is shattered into all samples where each sample only gives $d$ out of $m$ coordinates of $(p - q)$, and there is no succinct representation for $\mathrm{Cov}[f(X, q)]$.

The second difficulty is that $f(X, q)$ is $m$-dimensional. We cannot explicitly write down all the samples $\{f(X_i, q)\}_{i=1}^N$, because this takes time $\Omega(Nm)$, which could be much slower than our desired running time of $\widetilde{O}(Nd)$. Similarly, a black-box use of nearly-linear time robust mean estimation algorithms (e.g., Lemma 2.6) runs in time $\Omega(Nm)$, which is too slow.

In the rest of this section, we explain how we handle these two issues.

**Stability Condition of $f(X, q)$.**  Because the second-order stability condition in Lemma 2.3 is defined with respect to $I$, we first scale the samples so that the covariance of $f(X, p)$ becomes $I$. Lemma 2.3 shows that $\mathrm{Cov}[f(X, p)] = \mathrm{diag}(\pi^P \circ p \circ (1 - p))$. To make it close to $I$, we can multiply the $k$-th coordinate of $f(X, p)$ by $(\pi_k^P p_k (1 - p_k))^{-1/2}$. However, we do not know the exact value of $\pi^P$ or $p$, instead we use the corresponding empirical estimates $\pi^S$ and $q^S$ (see Algorithm 1).

**Definition 3.1.** *Let $\pi^S$ and $q^S$ denote the parental configuration probabilities and conditional means estimated over $S$. Let $s = 1/\sqrt{(\pi^S \circ q^S \circ (1 - q^S))}$. Throughout this paper, for a vector $v \in \mathbb{R}^m$, we use $\hat{v} \in \mathbb{R}^m$ to denote $v \circ s$. In particular, we have $\hat{X}_i = X_i \circ s$ (and similarly $\hat{p}$, $\hat{q}$, $\hat{f}(x, q)$).*

Now we analyze the concentration bounds for $\hat{f}(X, q)$. Formally, we prove the following lemma.

**Lemma 3.2.** *Assume the conditions in Section 2.3 hold for the original set of good samples $G^\star$. Then, for $\delta_1 = \epsilon \sqrt{\log 1/\epsilon}$ and $\delta_2 = \epsilon \log(1/\epsilon)$, the set of samples $\{\hat{f}(X_i, q)\}_{i \in G^\star}$ is*

$$\left(\epsilon, O\left(\frac{\delta_1}{\sqrt{\alpha c}} + \epsilon \|\hat{p} - \hat{q}\|_2\right), O\left(\frac{\delta_2}{\alpha c} + B + \sqrt{B}\right)\right)\text{-stable,}$$

*where $B = \|\sqrt{\pi^P} \circ (\hat{p} - \hat{q})\|_2^2$.*

We provide some intuition for Lemma 3.2 and defer its proof to Appendix B.

For the first moment, the difference between $\mathbb{E}[\hat{f}(X, q)]$ and the empirical mean of $\hat{f}(X, q)$ comes from several places. Even if $q = p$, we would incur an error of $\delta_1$ from the concentration bound in Equation equation 3, which is at most $\delta_1(\alpha c)^{-1/2}$ after the scaling by $s$. Moreover, on average $\pi_k^P$ fraction of the samples gives us information about $(\hat{p} - \hat{q})_k$. Since an $\epsilon$-fraction of the samples are removed when proving stability, we may only have $(\pi_k^P - \epsilon)$-fraction instead, which introduces an error of $\epsilon \|\hat{p} - \hat{q}\|_2$. This is why the first-moment parameter is $(\delta_1(\alpha c)^{-1/2} + \epsilon \|\hat{p} - \hat{q}\|_2)$.

For the second moment, after the scaling, we have $\mathrm{Cov}[\hat{f}(X, p)] \approx I$. Ideally, we would like to prove $\mathrm{Cov}[\hat{f}(X, q)] \approx I + (\pi^P \circ (\hat{p} - \hat{q}))(\pi^P \circ (\hat{p} - \hat{q}))^\top$, but this is too good to be true. For two coordinates $k \neq \ell$, whether a sample gives information about $(\hat{p} - \hat{q})_k$ or $(\hat{p} - \hat{q})_\ell$ is not independent. We can upper bound the probability that both parental configurations happen by $\min(\pi_k^P, \pi_\ell^P)$. If they were independent we would have a bound of $\pi_k^P \pi_\ell^P$. The difference in these two upper bounds is intuitively why $\sqrt{\pi^P}$ appears in the second-moment parameter. See Appendix B for more details.

**Robust Mean Estimation with Sparse Input.**  To overcome the second difficulty, we exploit the sparsity of the expanded vectors. Observe that each vector $f(X, q)$ is guaranteed to be $d$-sparse because exactly $d$ parental configuration can happen (see Definition 2.2). The same is true for $\hat{f}(X, q)$ because scaling does not change the number of nonzeros. Therefore, there are in total $O(Nd)$ nonzero entries in the set of samples $\{\hat{f}(X, q)\}_{i \in S}$.

We develop a robust mean estimation algorithm that runs in time nearly-linear in the *number of nonzeros* in the input. Combined with the above argument, if we only invoke this mean estimation algorithm polylogarithmic times, we can get the desired running time of $\widetilde{O}(Nd)$.

**Lemma 3.3.** *Consider the same setting as in Lemma 2.6. Suppose $\|X\|_2 \leq R$ for all $X \in S$. There is an algorithm $\mathscr{A}_{mean}$ with the same error guarantee that runs in time $\widetilde{O}(\log R \cdot (\mathrm{nnz}(S) + N + d))$ where $\mathrm{nnz}(S)$ is the total number of nonzeros in $S$. That is, given an $\epsilon$-corrupted version of an $(\epsilon, \beta, \gamma)$-stable set of $N$ samples w.r.t. a $d$-dimensional distribution with mean $\mu_X$, the algorithm $\mathscr{A}_{mean}$ outputs an estimator $\hat{\mu} \in \mathbb{R}^d$ in time $\widetilde{O}(\log R \cdot (\mathrm{nnz}(S) + N + d))$ such that with high probability, $\|\hat{\mu} - \mu_X\|_2 \leq O(\sqrt{\epsilon\gamma} + \beta + \epsilon\sqrt{\log 1/\epsilon})$ .*

We prove Lemma 3.3 by extending the algorithm in Dong et al. (2019) to handle sparse input. The main computation bottleneck of recent nearly-linear time robust mean estimation algorithms (Cheng et al., 2018; Dong et al., 2019) is in using the matrix multiplicative weight update (MMWU) method. In each iteration of MMWU, a score is computed for each sample. Roughly speaking, this score indicates whether one should continue to increase the weight on the corresponding sample. Previous algorithms use the Johnson-Lindenstrauss lemma to approximate the scores for all $N$ samples simultaneously. We show that the sparsity of the input vectors allows for faster application of the Johnson-Lindenstrauss lemma, and all $N$ scores can be computed in time nearly-linear in $\mathrm{nnz}(S)$.

We defer the proof of Lemma 3.3 to Appendix C.

## 4 ROBUST LEARNING OF BAYESIAN NETWORKS IN NEARLY-LINEAR TIME

In this section, we prove our main result. We present our algorithm (Algorithm 1) and prove its correctness and analyze its running time (Theorem 4.1).

**Theorem 4.1.** *Fix $0 < \epsilon < \epsilon_0$ where $\epsilon_0$ is a sufficiently small universal constant. Let $P$ be a $c$-balanced Bayesian network on $\{0, 1\}^d$ with known structure $H$. Let $\alpha$ be the minimum parental configuration probability of $P$. Assume $\alpha = \widetilde{\Omega}(\epsilon^{2/3}c^{-1/3})$.*

*Let $S$ be an $\epsilon$-corrupted set of $N = \widetilde{\Omega}(m/\epsilon^2)$ samples drawn from $P$. Given $H$, $S$, $\epsilon$, $c$, and $\alpha$, Algorithm 1 outputs a Bayesian network $Q$ in time $\widetilde{O}(Nd)$ such that, with probability at least $9/10$, $d_{\mathrm{TV}}(P, Q) \leq O(\epsilon\sqrt{\log(1/\epsilon)}/\sqrt{\alpha c})$.*

The $c$ and $\alpha$ terms in the error guarantee also appear in prior work (Cheng et al., 2018). Removing this dependence is an important technical question that is beyond the scope of this paper.

Theorem 4.1 follows from three key technical lemmas. At the beginning of Algorithm 1, we first scale all the input vectors as in Definition 3.1. We maintain a guess $q$ for $p$ and gradually move it closer to $p$. In our analysis, we track our progress by the $\ell_2$-norm of $\pi^P \circ (\hat{p} - \hat{q})$.

Initially, we set $q$ to be the empirical conditional mean over $S$. Lemma 4.2 proves that $\|\pi^P \circ (\hat{p} - \hat{q}')\|_2$ is not too large for our first guess. Lemma 4.3 shows that, as long as $q$ is still relatively far from $p$, we can compute a new guess such that $\|\pi^P \circ (\hat{p} - \hat{q})\|_2$ decreases by a constant factor. Lemma 4.4 states that, when the algorithm terminates and $\|\pi^P \circ (\hat{p} - \hat{q})\|_2$ is small, we can conclude that the output $Q$ is close to the ground-truth $P$ in total variation distance.

In the following three lemmas, we consider the same setting as in Theorem 4.1 and assume the conditions in Section 2.3 hold.

**Lemma 4.2** (Initialization). *In Algorithm 1, we have*

$$\left\|\pi^P \circ (\hat{p} - \hat{q}^0)\right\|_2 \leq O(\epsilon\sqrt{d}/\sqrt{\alpha c}) .$$

**Lemma 4.3** (Iterative Refinement). *Fix an iteration $t$ in Algorithm 1. Assume the robust mean estimation algorithm $\mathscr{A}_{mean}$ succeeds. If $\left\|\pi^P \circ (\hat{p} - \hat{q}^t)\right\|_2 \leq \rho^t$ and $\rho^t = \Omega(\epsilon\sqrt{\log(1/\epsilon)}/\sqrt{\alpha c})$, then we have $\left\|\pi^P \circ (\hat{p} - \hat{q}^{t+1})\right\|_2 \leq c_1\rho^t$ for some universal constant $c_1 < 1$.*

**Lemma 4.4.** *Let $Q$ be a Bayesian network that has the same structure as $P$. Suppose that (1) $P$ is $c$-balanced, (2) $\alpha = \Omega(r + \epsilon/c)$, and (3) $\left\|\pi^P \circ (\hat{p} - \hat{q})\right\|_2 \leq r/2$. Then we have $d_{\mathrm{TV}}(P, Q) \leq r$.*

We defer the proofs of Lemmas 4.2, 4.3, and 4.4 to Appendix D and we first prove Theorem 4.1.

*Proof of Theorem 4.1.* We first prove the correctness of Algorithm 1.

---

**Algorithm 1:** Robustly Learning Bayesian Networks

---

**Input** : The dependency graph $H$ of a $c$-balanced Bayesian network $P$ with minimum
parental configuration $\alpha$, an $\epsilon$-corrupted set $S$ of $N = \widetilde{\Omega}(m/\epsilon^2)$ samples $\{X_i\}_{i=1}^N$
drawn from $P$, and the values of $\epsilon$, $c$ and $\alpha$.

**Output:** A Bayesian network $Q$ such that, with probability at least $9/10$,
$\quad d_{\mathrm{TV}}(P, Q) \leq O(\epsilon\sqrt{\log(1/\epsilon)}/\sqrt{\alpha c})$.

Compute the empirical probabilities $\pi^S$ where $\pi^S(i, a) = \Pr_{X \in S}[\Pi_{i,a}]$;

Compute the empirical conditional probabilities $q^S$ where $q^S(i, a) = \Pr_{X \in S}[X(i) = 1 \mid \Pi_{i,a}]$;

Compute the scaling vector $s = 1/\sqrt{(\pi^S \circ q^S \circ (1 - q^S))}$;

Let $T = O(\log d)$ and $q^0 = q^S$;

Let $\rho^0 = O(\epsilon\sqrt{d}/\sqrt{\alpha c})$. (We maintain upper bounds $\rho^t$ s.t. $\left\|\pi^P \circ (\hat{p} - \hat{q}^t)\right\|_2 \leq \rho^t$ for all $t$);

**for** $t = 0$ **to** $T - 1$ **do**
$\quad$ $\beta^t = O(\epsilon\rho^t/\alpha), \gamma^t = O((\rho^t)^2/\alpha + \rho^t/\sqrt{\alpha})$ ;
$\quad$ Solve a robust mean estimation problem. Let $\nu = \mathscr{A}_{mean}\left(\{\hat{f}(X_i, q^t)_{i \in S}\}, \epsilon, \beta^t, \gamma^t\right)$;
$\quad$ $q^{t+1} = \nu \circ (1/s) \circ (1/\pi^S) + q^t; \qquad \rho^{t+1} = c_1\rho^t$;

**return** *the Bayesian network $Q$ with graph $H$ and conditional probabilities $q^T$* ;

---

The original set of $N = \Omega(m \log(m/\epsilon)/\epsilon^2)$ good samples drawn from $P$ satisfies the conditions in Section 2.3 with probability at least $1 - \frac{1}{20}$. With high probability, the robust mean estimation oracle $\mathscr{A}_{mean}$ succeeds in all iterations. For the rest of this proof, we assume the above conditions hold, which by a union bound happens with probability at least $9/10$.

From Lemma 4.2, we have the following condition on the initial estimate $q^0$.

$$\left\|\pi^P \circ (\hat{p} - \hat{q}^0)\right\|_2 = O(\epsilon\sqrt{d}/\sqrt{\alpha c}) \ .$$

We start with an upper bound $\rho^0$ of $\left\|\pi^P \circ (\hat{p} - \hat{q}^0)\right\|_2 = O(\epsilon\sqrt{d}/\sqrt{\alpha c})$. By Lemma 4.3, in each iteration, if $\rho^t = \Omega(\epsilon\sqrt{\log(1/\epsilon)}/\sqrt{\alpha c})$, we can obtain a new estimate $q^{t+1}$ and an upper bound $\rho^{t+1}$ on $\left\|\pi^P \circ (\hat{p} - \hat{q}^t)\right\|_2$ such that $\rho^{t+1}$ is smaller than $\rho^t$ by a constant factor. Hence after $O(\log(d))$ iterations, we can get a vector $q^t$ such that

$$\left\|\pi^P \circ (\hat{p} - \hat{q}^t)\right\|_2 = O(\epsilon\sqrt{\log(1/\epsilon)}/\sqrt{\alpha c}) \ .$$

Let $Q$ be the Bayesian network with conditional probability table $q^t$. The assumption that $\alpha = \widetilde{\Omega}(\epsilon^{2/3}c^{-1/3})$ allows us to apply Lemma 4.4 with $r = O(\epsilon\sqrt{\log(1/\epsilon)}/\sqrt{\alpha c})$, which gives the claimed upper bound on $d_{\mathrm{TV}}(P, Q)$.

Now we analyze the runtime of Algorithm 1. First, $q^S$ and $\pi^S$ can be computed in time $\widetilde{O}(Nd)$ because each sample only affects $d$ entries of $q$. We do not explicitly write down $f(X, q)$. In each iteration, we solve a robust mean estimation problem with input $\left\{\hat{f}(X_i, q^t)\right\}_{i \in S}$, which takes time $\widetilde{O}(Nd)$. This is because there are $N$ input vectors, each vector is $d$-sparse, and the robust mean estimation algorithm runs in time nearly-linear in the number of nonzeros in the input (Lemma 3.3). We can compute $q^{t+1} = \nu \circ (1/s) \circ (1/\pi^S) + q^t$ in time in time $O(m)$.

Since there are $O(\log d)$ iterations, the overall running time is

$$\widetilde{O}(Nd) + O(\log d)\left(\widetilde{O}(Nd) + O(m)\right) = \widetilde{O}(Nd) \ . \qquad \qquad \square$$

ACKNOWLEDGMENTS

Part of this work was done while Yu Cheng was visiting the Institute of Advanced Study. Part of this work was done while Honghao Lin was an undergraduate student at Shanghai Jiao Tong University.

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

## A   DETERMINISTIC CONDITIONS ON GOOD SAMPLES

In this section, we will first prove Lemma 2.3, then we prove that the deterministic conditions in Section 2.3 hold with high probability if we take enough samples.

**Lemma A.1.** *For $X \sim P$ and $f(X, p)$ as defined in Definition 2.2, we have*

(i) $\mathbb{E}(f(X, p)) = 0$.

(ii) $\mathrm{Cov}[f(X, p)] = \mathrm{diag}(\pi^P \circ p \circ (1 - p))$.

(iii) *For any unit vector $v \in \mathbb{R}^m$, we have $\mathrm{Pr}_{X \sim P}\left[|v^\top f(X, p)| \geq T\right] \leq 2 \exp(-T^2/2)$.*

*Proof.* We first claim that $\mathbb{E}_{X \sim P}[f(X, p)_k | f(X, p)_1, ..., f(X, p)_{k-1}] = 0$ for all $k \in [m]$. Let $k = (i, a)$, conditioned on $f(X, p)_1, ..., f(X, p)_{k-1}$, the event $\pi_{i,a}$ may or may not happen. A simple calculation shows that in both cases, we have $\mathbb{E}_{X \sim P}[f(X, p)_k | f(X, p)_1, ..., f(X, p)_{k-1}] = 0$.

For $(i)$, we have $\mathbb{E}[f(X, p)] = \pi^P \circ p + (1 - \pi^P) \circ p - p = 0$.

For $(ii)$, we first show that for any $(i, a) \neq (j, b)$, we have $\mathbb{E}[f(X, p)_{i,a} f(X, p)_{j,b}] = 0$. For the case $i = j$, we can see at least one of $\Pi_{i,a}$ and $\Pi_{j,b}$ does not occur, so $f(X, p)_{i,a} f(X, p)_{j,b}$ is always 0. For the case $i \neq j$, we assume without loss of generality that $i > j$, then we have $\mathbb{E}[f(X, p)_{i,a} | f(X, p)_{j,b}] = 0$.

For all $(i, a) \in [m]$, we have $\mathbb{E}[f(X, p)_{i,a}^2] = \pi_{i,a}^P \mathbb{E}[(X - p_{i,a})^2 | \Pi_{i,a}] = \pi_{i,a}^P p_{i,a}(1 - p_{i,a})$. Combining these two, we get $\mathrm{Cov}[f(X, p)] = \mathrm{diag}(\pi^P \circ p \circ (1 - p))$.

For $(iii)$, we recall that $\mathbb{E}_{X \sim P}[f(X, p)_k | f(X, p)_1, ..., f(X, p)_{k-1}] = 0$, thus the sequence $\sum_{k=1}^{\ell} v_k f(X, q)_k$ for $1 \leq \ell \leq m$ is a martingale, and we can apply Azuma's inequality. Note that $|v_k| \geq |v_k f(X, p)_k|$, hence we have $\mathrm{Pr}_{X \sim P}\left[|v^\top f(X, p)| \geq T\right] \leq 2 \exp(-T^2/2 \|v\|_2^2) = 2 \exp(-T^2/2)$. $\qquad \square$

The conditions in Equations equation 1 and equation 2 are proved in Lemma A.2, and the conditions in Equation equation 3 are proved in Corollary A.4.

**Lemma A.2.** *Let $P$ be a Bayesian network. Let $G^\star$ be a set of $\Omega((m \log(m/\tau))/\epsilon^2)$ samples drawn from $P$. Let $\pi^{G^\star}$ and $p^{G^\star}$ be the empirical parental configuration probabilities and conditional probabilities of $P$ given by $G^\star$. Then with probability $1 - \tau$, the following conditions hold:*

(i) *For any subset $T \subset G^\star$ with $|T| \geq (1 - 2\epsilon)N$, we have*

$$\left\|\pi^T - \pi^P\right\|_\infty \leq O(\epsilon) .$$

(ii)

$$\left\|\sqrt{\pi^P} \circ (p - p^{G^\star})\right\|_2 \leq O(\epsilon) .$$

*Proof.* For $(i)$, first consider the case of $T = G^\star$ and fix an entry $1 \leq k \leq m$ in the conditional probability table. Because each sample is drawn independently from $P$, by the Chernoff bound, we have that when $N = \Omega(\log(m/\tau)/\epsilon^2)$, $|\pi_k^P - \pi_k^{G^\star}| \leq \epsilon$ holds with probability at least $1 - \tau/m$. Hence, after taking an union bound over $k$, we have that $\left\|\pi^T - \pi^P\right\|_\infty \leq \epsilon$ holds with probability at least $1 - \tau$. Now for a general subset $T \subset G^\star$, notice that removing $O(\epsilon)$-fraction of samples can change $\pi^T$ by at most $O(\epsilon)$. Thus, condition $(i)$ holds with probability at least $1 - \tau$.

For $(ii)$, for any $k = (i, a)$, note that $p_k^{G^\star}$ is estimated from $\pi_k^{G^\star} N$ samples. In these samples, the parental configuration $\Pi_k$ happens and the value of $X_i$ is decided independently. By the Chernoff bound and the union bound, we get that when $N = \Omega((m \log(m/\tau))/\epsilon^2)$, $|p_k^{G^\star} - p_k| \leq \epsilon/\sqrt{m \pi_k^{G^\star}}$ holds for every $k$ with probability at least $1 - \tau$, which implies

$$\left\|\sqrt{\pi^{G^\star}} \circ (p - p^{G^\star})\right\|_2 \leq O(\epsilon).$$

Combining this with $\left\|\pi^P - \pi^{G^\star}\right\|_\infty \leq \epsilon$, we get that condition $(ii)$ holds. $\qquad \square$

To prove Equation equation 3, we use the following concentration bounds for subgaussian distributions. Recall that a distribution $D$ on $\mathbb{R}^d$ with mean $\mu$ is subgaussian if for any unit vector $v \in \mathbb{R}^d$ we have $\Pr_{x \sim D}[|\langle v, x - \mu \rangle| \geq t] \leq \exp(-ct^2)$, where $c$ is a universal constant.

**Lemma A.3.** *Let $G^\star$ be a set of $N = \Omega((\epsilon\sqrt{\log 1/\epsilon})^{-2}(d + \log(1/\tau)))$ samples drawn from a d-dimensional subgaussian distribution with mean $\mu$ and covariance matrix $\Sigma \preceq I$. Here $A \preceq B$ means that $B - A$ is a positive semi-definite matrix. Then, with probability $1 - \tau$, the following conditions hold:*

*For $\delta_1 = c_1(\epsilon\sqrt{\log 1/\epsilon})$ and $\delta_2 = c_1(\epsilon \log 1/\epsilon)$ where $c_1$ is an universal constant, we have that for any subset $T \subset G^\star$ with $|T| \geq (1 - 2\epsilon)N$,*

$$\left\| \frac{1}{|T|} \sum_{i \in T} (X_i - \mu) \right\|_2 \leq \delta_1 , \qquad \left\| \frac{1}{|T|} \sum_{i \in T} (X_i - \mu)(X_i - \mu)^\top - \Sigma \right\|_2 \leq \delta_2 \qquad (4)$$

A special case of Lemma A.3 where $\Sigma = I$ is proved in Diakonikolas et al. (2019a). The proof for the general case where $\Sigma \preceq I$ is almost identical. In particular, the concentration inequalities used in Diakonikolas et al. (2019a) for subgaussian distributions still hold when $\Sigma \preceq I$ (see, e.g., Vershynin (2010)).

From Lemma A.3 and 2.3, we have the following corollary:

**Corollary A.4.** *Let $G^\star$ be a set of $N = \Omega((\epsilon\sqrt{\log 1/\epsilon})^{-2}(m + \log(1/\tau)))$ samples drawn $P$. Then, with probability $1 - \tau$, the following conditions to hold: For $\delta_1 = c_1(\epsilon\sqrt{\log 1/\epsilon})$ and $\delta_2 = c_1(\epsilon \log 1/\epsilon)$, where $c_1$ is an universal constant, we have that for any subset $T \subset G^\star$ with $|T| \geq (1 - 2\epsilon)N$,*

$$\left\| \frac{1}{|T|} \sum_{i \in T} f(X_i, p) \right\|_2 \leq O(\delta_1) , \qquad \left\| \frac{1}{|T|} \sum_{i \in T} f(X_i, p)f(X_i, p)^\top - \Sigma \right\|_2 \leq O(\delta_2) , \qquad (5)$$

*where $\Sigma = \mathrm{Cov}[f(X, p)] = \mathrm{diag}(\pi^P \circ p \circ (1 - p))$.*

# B  STABILITY CONDITION OF $\hat{f}(X, q)$

In this section, we prove the stability condition for the samples $\hat{f}(X, q)$ (Lemma 3.2). Recall the definition of $\hat{f}(X, q)$ from Definitions 2.2 and 3.1. We first restate Lemma 3.2.

**Lemma 3.2.** *Assume the conditions in Section 2.3 hold for the original set of good samples $G^\star$. Then, for $\delta_1 = \epsilon\sqrt{\log 1/\epsilon}$ and $\delta_2 = \epsilon \log(1/\epsilon)$, the set of samples $\{\hat{f}(X_i, q)\}_{i \in G^\star}$ is*

$$\left( \epsilon, O\Big( \frac{\delta_1}{\sqrt{\alpha c}} + \epsilon \|\hat{p} - \hat{q}\|_2 \Big), O\Big( \frac{\delta_2}{\alpha c} + B + \sqrt{B} \Big) \right) \text{-stable,}$$

*where $B = \|\sqrt{\pi^P} \circ (\hat{p} - \hat{q})\|_2^2$.*

We will prove the stability of $f(X, q)$. The stability of $\hat{f}(X, q)$ follows directly. We introduce a matrix $C_{D,q}$ which is crucial in proving the stability of $f(X, q)$. Intuitively, the matrix $C_{D,q}$ is related to the difference in the covariance of $f(X, p)$ and that of $f(X, q)$ on the sample set $D$.

**Definition B.1.** *For any set $D$ of samples $\{X_i\}_{i \in D}$, we define the following $m \times m$ matrix*

$$C_{D,q} = \frac{1}{|D|} \sum_{i \in D} (f(X_i, p) - f(X_i, q))(f(X_i, p) - f(X_i, q))^\top .$$

Observe that for $x \in \{0, 1\}^d$ with $x \notin \Pi_k$, we have $f(x, p)_k = f(x, q)_k = 0$. On the other hand, if $x \in \Pi_k$ for some $k = (i, a)$, we have $f(x, p)_k - f(x, q)_k = (x_i - p_k) - (x_i - q_k) = q_k - p_k$.

In the very special case where all parental configurations happen (i.e., a binary product distribution), we would have $C_{D,q} = (p - q)(p - q)^\top$. However, in general the information related to $(p - q)$ is spread among the samples. We show that even though $C_{D,q}$ does not have a succinct representation, we can prove the following upper bound on the spectral norm of $C_{D,q}$.

**Lemma B.2.** $\|C_{D,q}\|_2 \le \sum_k \pi_k^D (p_k - q_k)^2$ .

*Proof.* For notational convenience, let $C = C_{D,q}$. For every $1 \le k, \ell \le m$, we have

$$|C_{k,\ell}| = \left| (\Pr_D[\Pi_k \wedge \Pi_\ell])(p_k - q_k)(p_\ell - q_\ell) \right| \le \min\{\pi_k^D, \pi_\ell^D\} \cdot |(p_k - q_k)(p_\ell - q_\ell)|$$
$$\le \left( \sqrt{\pi_k^D} |(p_k - q_k)| \right) \cdot \left( \sqrt{\pi_\ell^D} |(p_\ell - q_\ell)| \right)$$

We can upper bound the spectral norm of $C$ in term of its Frobenius norm:

$$\|C\|_2^2 \le \|C\|_F^2 = \sum_{k,\ell} C_{k,\ell}^2 \le \sum_{k,\ell} \left( \pi_k^D (p_k - q_k)^2 \right) \left( \pi_\ell^D (p_\ell - q_\ell)^2 \right) \le \left( \sum_k \pi_k^D (p_k - q_k)^2 \right)^2 .$$

$\square$

The following lemma essentially proves the stability of $f(X, q)$, except that in the second-order condition, we should have $\mathrm{Cov}(f(X, q))$ instead of $\Sigma$. We will bridge this gap in Lemma B.4.

**Lemma B.3.** *Assume the conditions in Section 2.3 hold. For $\delta_1 = \epsilon\sqrt{\log 1/\epsilon}$ and $\delta_2 = \epsilon \log(1/\epsilon)$, we have that for any subset $T \subset G^\star$ with $|T| \ge (1 - \epsilon)|G^\star|$,*

$$\left\| \frac{1}{|T|} \sum_{i \in T} (f(X_i, q) - \pi^P \circ (p - q)) \right\|_2 \le O(\delta_1 + \epsilon \|p - q\|_2) , \text{ and}$$

$$\left\| \frac{1}{|T|} \sum_{i \in T} (f(X_i, q) - \pi^P \circ (p - q))(f(X_i, q) - \pi^P \circ (p - q))^\top - \Sigma \right\|_2 \le O(\delta_2 + B + \sqrt{B})$$

*where $B = \left\| \sqrt{\pi^P} \circ (p - q) \right\|_2^2 \le \frac{1}{\alpha} \left\| \pi^P \circ (p - q) \right\|_2^2$, and $\Sigma = \mathrm{diag}(\pi^P \circ p \circ (1 - p))$ is the true covariance of $f(X, p)$ .*

*Proof.* For the first moment, we have

$$\left\| \frac{1}{|T|} \sum_{i \in T} (f(X_i, q) - \pi^P \circ (p - q)) \right\|_2$$
$$= \left\| \frac{1}{|T|} \sum_{i \in T} (f(X_i, p) + f(X_i, q) - f(X_i, p) - \pi^P \circ (p - q)) \right\|_2$$
$$\le \left\| \frac{1}{|T|} \sum_{i \in T} f(X_i, p) \right\|_2 + \left\| \frac{1}{|T|} \sum_{i \in T} (f(X_i, q) - f(X_i, p) - \pi^P (p - q)) \right\|_2$$
$$= \left\| \frac{1}{|T|} \sum_{i \in T} f(X_i, p) \right\|_2 + \left\| (\pi^T - \pi^P) \circ (p - q) \right\|_2 = O(\delta_1 + \epsilon \|p - q\|_2) .$$

For the second moment, consider any unit vector $v \in \mathbb{R}^m$. We have

$$v^\top \left( \frac{1}{|T|} \sum_{i \in T} f(X_i, q) f(X_i, q)^\top \right) v = \frac{1}{|T|} \sum_{i \in T} \langle f(X_i, q), v \rangle^2$$
$$= \frac{1}{|T|} \sum_{i \in T} \left( \langle f(X_i, p), v \rangle^2 + \langle f(X_i, p) - f(X_i, q), v \rangle^2 + 2\langle f(X_i, p), v \rangle \langle f(X_i, p) - f(X_i, q), v \rangle \right)$$
$$\le \frac{1}{|T|} \sum_{i \in T} \left( \langle f(X_i, p), v \rangle^2 + \langle f(X_i, p) - f(X_i, q), v \rangle^2 \right)$$
$$+ 2\sqrt{\frac{1}{|T|} \left( \sum_{i \in T} \langle f(X_i, p), v \rangle^2 \right) \left( \frac{1}{|T|} \sum_{i \in T} \langle f(X_i, p) - f(X_i, q), v \rangle^2 \right)}$$

where the last inequality follows from the Cauchy-Schwarz inequality. Therefore, we have

$$
\left\| \frac{1}{|T|} \sum_{i \in T} f(X_i, q) f(X_i, q)^\top - \Sigma \right\|_2
$$

$$
\leq \left\| \frac{1}{|T|} \sum_{i \in T} f(X_i, p) f(X_i, p)^\top - \Sigma \right\|_2
$$

$$
+ \left\| \frac{1}{|T|} \sum_{i \in T} (f(X_i, p) - f(X_i, q))(f(X_i, p) - f(X_i, q))^\top \right\|_2
$$

$$
+ 2 \sqrt{ \left\| \frac{1}{|T|} \sum_{i \in T} f(X_i, p) f(X_i, p)^\top \right\|_2 \left\| \frac{1}{|T|} \sum_{i \in T} (f(X_i, p) - f(X_i, q))(f(X_i, p) - f(X_i, q))^\top \right\|_2 }
$$

$$
\leq \delta_2 + \|C_{T,q}\|_2 + 2\sqrt{1 + \delta_2} \sqrt{\|C_{T,q}\|_2} = O\left( \delta_2 + \|C_{T,q}\|_2 + \sqrt{\|C_{T,q}\|_2} \right)
$$

Finally, we show that the second moment matrix $\frac{1}{|T|} \sum_{i \in T} f(X_i, q) f(X_i, q)^\top$ is not too far from the empirical covariance matrix of $f(X, q)$.

$$
\left\| \frac{1}{|T|} \sum_{i \in T} f(X_i, q) f(X_i, q)^\top - \frac{1}{|T|} \sum_{i \in T} (f(X_i, q) - \pi^P \circ (p - q))(f(X_i, q) - \pi^P \circ (p - q))^\top \right\|_2
$$

$$
\leq 2 \left\| \frac{1}{|T|} \sum_{i \in T} (f(X_i, q) - \pi^P \circ (p - q)) \right\|_2 \|\pi_P \circ (p - q)\|_2 + \|\pi_P \circ (p - q)\|_2^2
$$

$$
\leq O\left( (\delta_1 + \epsilon \|p - q\|_2) \|\pi_P \circ (p - q)\|_2 \right) \leq O\left( \|\pi_P \circ (p - q)\|_2^2 \right) .
$$

Putting everything together and using Lemma B.2, we conclude this proof. $\qquad\square$

The stability of $\hat{f}(X, q)$ follows from the stability of $f(X, q)$ (Lemma B.3), scaling all samples by $s$, and replacing $\hat{\Sigma}$ with $I$ in the second-order condition using Lemma B.4.

**Lemma B.4.** *Assume the conditions in Section 2.3 hold. Then after scaling, we have*

$$
\left\| \hat{\Sigma} - I \right\|_2 \leq O(\frac{\epsilon}{\alpha c}) .
$$

*where $\hat{\Sigma}$ is the covariance matrix of $\hat{f}(X, p)$.*

*Proof.* We recall that $\pi_k^P p_k (1 - p_k) \geq \frac{1}{2} \pi_k^P \min(p_k, 1 - p_k) = \Omega(\alpha c)$. Because $\|s\|_\infty = O(1/\sqrt{\alpha c})$, it suffices to show that

$$
\|\mathrm{Cov}(s \circ f(X, p)) - I\|_2 \leq O(\epsilon) .
$$

In other words, we need to show

$$
\left\| \pi^P \circ p \circ (1 - p) - \pi^S \circ q^S \circ (1 - q^S) \right\|_\infty = O(\epsilon) .
$$

Let $\pi^{G^\star}$ and $p^{G^\star}$ be the empirical parental configuration probabilities and conditional probabilities of $P$ given by $G^\star$. We first prove that

$$
\left\| \pi^{G^\star} \circ p^{G^\star} \circ (1 - p^{G^\star}) - \pi^S \circ q^S \circ (1 - q^S) \right\|_\infty = O(\epsilon) .
$$

Note that $\left\| \pi^{G^\star} - \pi^S \right\|_\infty = O(\epsilon)$ and $q_k^S (1 - q_k^S) < 1$, so it is sufficient to show that

$$
\left\| \pi^{G^\star} \circ p^{G^\star} \circ (1 - p^{G^\star}) - \pi^{G^\star} \circ q^S \circ (1 - q^S) \right\|_\infty = O(\epsilon) .
$$

We have

$$\left\| \pi^{G^\star} \circ p^{G^\star} \circ (1 - p^{G^\star}) - \pi^{G^\star} \circ q^S \circ (1 - q^S) \right\|_\infty$$
$$= \left\| \pi^{G^\star} \circ p^{G^\star} - \pi^{G^\star} \circ q^S - \pi^{G^\star} \circ (p^{G^\star} + q^S) \circ (p^{G^\star} - q^S) \right\|_\infty \le 3 \left\| \pi^{G^\star} \circ (p^{G^\star} - q^S) \right\|_\infty .$$

Let $n_k$ denote the number of times that the event $\Pi_k$ happens over $G^\star$, and let $t_k$ be the number of times that $X(i) = 1$ when $\Pi_k = \Pi_{i,a}$ happens. Because $S$ is obtained by changing at most $\epsilon N$ samples in $G^\star$, we can get

$$|\pi_k^{G^\star} \circ (p_k^{G^\star} - q_k^S)| \le \frac{n_k}{N} \cdot \left( \frac{t_k + \epsilon N}{n_k - \epsilon N} - \frac{t_k}{n_k} \right) = \frac{n_k(n_k + t_k)\epsilon N}{N n_k(n_k - \epsilon N)} \le \frac{2 n_k \epsilon N}{0.5 N n_k} = 4\epsilon .$$

The last inequality follows from $t_k \le n_k$ and $n_k - \epsilon N \ge 0.5 n_k$ (because we assume the minimum parental configuration probability is $\Omega(\epsilon)$).

This concludes $\left\| \pi^{G^\star} \circ p^{G^\star} \circ (1 - p^{G^\star}) - \pi^S \circ q^S \circ (1 - q^S) \right\|_\infty = O(\epsilon)$.

Similarly, in order to prove that

$$\left\| \pi^P \circ p \circ (1 - p) - \pi^{G^\star} \circ p^{G^\star} \circ (1 - p^{G^\star}) \right\|_\infty = O(\epsilon) .$$

We just need to show $\left\| \pi^P \circ (p - p^{G^\star}) \right\|_2 = O(\epsilon)$, which is follows from $(ii)$ in Lemma A.2.

An application of triangle inequality finishes this proof. □

We are now ready to prove Lemma 3.2.

*Proof of Lemma 3.2.* By Lemma B.3 and the fact that $\|s\|_\infty = O(1/\sqrt{\alpha c})$, we know that for any subset $T \subset G^\star$ with $|T| \ge (1 - \epsilon)|G^\star|$, we have

$$\left\| \frac{1}{|T|} \sum_{i \in T} (\hat{f}(X_i, q) - \pi^P \circ (\hat{p} - \hat{q})) \right\|_2 \le O\left( \frac{\delta_1}{\sqrt{\alpha c}} + \epsilon \|\hat{p} - \hat{q}\|_2 \right) , \text{ and}$$

$$\left\| \frac{1}{|T|} \sum_{i \in T} (\hat{f}(X_i, q) - \pi^P \circ (\hat{p} - \hat{q}))(\hat{f}(X_i, q) - \pi^P \circ (\hat{p} - \hat{q}))^\top - \hat{\Sigma} \right\|_2 \le O\left( \frac{\delta_2}{\alpha c} + B + \sqrt{B} \right)$$

where $B = \left\| \sqrt{\pi^P} \circ (\hat{p} - \hat{q}) \right\|_2^2$, and $\hat{\Sigma}$ is the true covariance of $\hat{f}(X, p)$. This is because the scaling is applied to all vectors on both sides of the inequalities, so we only need to scale the scalars $\delta_1$ and $\delta_2$ appropriately.

We conclude the proof by replacing $\hat{\Sigma}$ in the second-order condition with $I$ using Lemma B.4. □

## C ROBUST MEAN ESTIMATION WITH SPARSE INPUT

In this section, we give a robust mean estimation algorithm that runs in nearly input-sparsity time. We build on the following lemma, which is essentially the main result of Dong et al. (2019).

**Lemma C.1** (essentially Dong et al. (2019)). *Given an $\epsilon$-corrupted version of an $(\epsilon, \beta, \gamma)$-stable set $S$ of $N$ samples w.r.t. a $d$-dimensional distribution with mean $\mu_X$. Suppose further that $\|X\|_2 \le R$ for all $X \in S$, there is an algorithm outputs an estimator $\hat{\mu} \in \mathbb{R}^d$ such that with high probability,*

$$\|\hat{\mu} - \mu_X\|_2 \le O(\sqrt{\epsilon\gamma} + \beta + \epsilon\sqrt{\log 1/\epsilon}) .$$

*Moreover, this algorithm runs in time $\widetilde{O}((\mathrm{nnz}(S) + N + d + T(\mathscr{O}_{apx})) \cdot \log R)$, where $\mathrm{nnz}(S)$ is the total number of nonzeros in the samples in $S$ and $T(\mathscr{O}_{apx}))$ is the runtime of an approximate score oracle defined in Definition C.2.*

Lemma C.1 is different from the way it is stated in Dong et al. (2019). This is because we use a more concise stability condition than the one in Dong et al. (2019). We will show that Lemma C.1 is equivalent to the version in Dong et al. (2019) in Appendix C.1.

The computational bottleneck of the algorithm in Dong et al. (2019) is logarithmic uses of matrix multiplicative weight update (MMWU). In each iteration of every MMWU, they need to compute a score for each sample. Intuitively, these scores help the algorithm decide whether it should continue to increase the weight on each sample or not.

We define some notations before we formally define the approximate score oracle. Let $\Delta_N = \{w \in \mathbb{R}^N : 0 \le w_i \le 1, \sum w_i = 1\}$ be the $N$-dimensional simplex. Given a set of $N$ samples $X_1, ..., X_N$ and a weight vector $w \in \Delta_N$, let $\mu(w) = \frac{1}{|w|} \sum w_i X_i$ and $\Sigma(w) = \frac{1}{|w|} \sum w_i (X_i - \mu(w))(X_i - \mu(w))^\top$ denote the empirical mean and covariance weighted by $w$.

**Definition C.2** (Approximate Score Oracle). *Given as input a set of $N$ samples $X_1, \ldots, X_N \in \mathbb{R}^d$, a sequence of $t + 1 = O(\log(d))$ weight vectors $w^0, \ldots, w^t \in \Delta_N$, and a parameter $\alpha > 0$, an approximate score oracle $\mathscr{O}_{apx}$ outputs $(1 \pm 0.1)$-approximations $(\widetilde{\tau}_i)_{i=1}^N$ to each of the $N$ scores*

$$\tau_i = (X_i - \mu(w^t))^\top U (X_i - \mu(w^t))$$

*for*

$$U = \frac{\exp(\alpha \sum_{i=0}^{t-1} \Sigma(w^i))}{\operatorname{tr} \exp(\alpha \sum_{i=0}^{t-1} \Sigma(w^i))} \ .$$

*In addition, $\mathscr{O}_{apx}$ outputs a scalar $\widetilde{q}$ such that*

$$|\widetilde{q} - q| \le 0.1q + 0.05 \left\| \Sigma(w^t) - I \right\|_2, \text{where } q = \langle \Sigma(w^t) - I, U \rangle \ .$$

These scores are computed using the Johnson-Lindenstrauss lemma. Our algorithm for computing these scores are given in Algorithm 2.

Let $r = O(\log N \log(1/\delta))$, $\ell = O(\log d)$, and $Q \in \mathbb{R}^{r \times d}$ be a matrix with i.i.d entries drawn from $\mathcal{N}(0, 1/r)$. Algorithm 2 computes an $r \times d$ matrix

$$A = Q \cdot P_\ell \left( \frac{\alpha}{2} \sum_{i=0}^{t-1} \Sigma(w^i) \right) \ . \tag{6}$$

where $P_\ell(Y) = \sum_{j=0}^{\ell} \frac{1}{j!} Y^j$ is a degree-$\ell$ Taylor approximation to $\exp(Y)$.

The estimates for the individual scores are then given by

$$\widetilde{\tau}_i = \frac{1}{\operatorname{tr}(AA^\top)} \left\| A(X_i - \mu(w^t)) \right\|_2 \tag{7}$$

and the estimate for $q$ is given by

$$\widetilde{q} = \sum_{i=1}^N (\widetilde{\tau}_i - 1) \ . \tag{8}$$

---

**Algorithm 2:** Nearly-linear time approximate score computation

---

**Input:** A set $S$ of $N$ samples $X_1, \ldots, X_N \in \mathbb{R}^d$, a sequence of weight vectors $w^0, ..., w^t$, a
    parameter $\alpha$, and a failure probability $\delta > 0$.
Let $r = O(\log N \log(1/\delta))$ and $\ell = O(\log d)$;
Let $Q \in \mathbb{R}^{r \times d}$ have entries drawn i.i.d. from $\mathcal{N}(0, 1/r)$;
Compute the matrix $A \in \mathbb{R}^{r \times d}$ as in Equation equation 6;
**return** $(\widetilde{\tau}_i)_{i=1}^N$ given by Equation equation 7 and $\widetilde{q}$ given by Equation equation 8;

---

The correctness of Algorithm 2 was proved in Dong et al. (2019).

**Lemma C.3** (Dong et al. (2019)). *With probability at least $1 - \delta$, the output of Algorithm 2 satisfies $|\widetilde{q} - q| \le 0.1q + 0.05 \|\Sigma(w^t) - I\|_2$ and $|\widetilde{\tau}_i - \tau_i| \le 0.1\tau_i$ for all $1 \le i \le N$.*

Consequently, we only need to analyze the runtime of Algorithm 2.

**Lemma C.4.** *Algorithm 2 runs in time $\widetilde{O}(t \cdot (N + d + \mathrm{nnz}(S)) \cdot \log(1/\delta))$.*

*Proof.* We first show that matrix $A \in \mathbb{R}^{r \times d}$ can be computed in time

$$\widetilde{O}(t \cdot (N + d + \mathrm{nnz}(S)) \cdot \log 1/\delta) .$$

We will multiply each row of $Q$ (from the left) through the matrix polynomial to obtain $A$. Let $v^\top \in \mathbb{R}^{1 \times d}$ be one of the rows of $Q$ and let $w \in \mathbb{R}^N$ be any weight vector. Observe that we can compute all $\left(v^\top (X_i - \mu(w))\right)_{i=1}^N$ in time

$$O\left(\sum_{i=1}^N \mathrm{nnz}(X_i) + N + d\right) = O(\mathrm{nnz}(S) + N + d) .$$

This is because we can compute $\mu(w)$ and $v^\top \mu(w)$ just once, and then compute $v^\top X_i$ for every $i$ and subtract $v^\top \mu(w)$ from it.

Then, we can compute

$$
\begin{aligned}
v^\top \Sigma(w) &= v^\top \left(\sum_{i=1}^N w_i(X_i - \mu(w))(X_i - \mu(w))^\top\right) \\
&= \sum_{i=1}^N w_i \left(v^\top(X_i - \mu(w))\right) X_i^\top - \left(\sum_{i=1}^N w_i \left(v^\top(X_i - \mu(w))\right)\right) \mu(w)^\top
\end{aligned}
$$

as the sum of $N$ sparse vectors subtracting a dense vector in time $O(\mathrm{nnz}(S) + N + d)$.

Therefore, for any $v \in \mathbb{R}^m$, we can evaluate $v^\top \sum_{i=0}^{t-1} \Sigma(w^i)$ in time $O(t \cdot (\mathrm{nnz}(S) + d + N))$.

Because $P_\ell$ is a degree-$\ell$ matrix polynomial of $\sum_{i=0}^{t-1} \Sigma(w^i)$, we can use Horner's method for polynomial evaluation to compute $v^\top P_\ell\left(-\frac{\alpha}{2} \sum_{i=0}^{t-1} \Sigma(w^i)\right)$ in time $O(\ell \cdot t \cdot (\mathrm{nnz}(S) + d + N))$. We need to multiply each of $r$ rows of $A$ through, we can compute $A$ in time $O(r \cdot \ell \cdot t \cdot (\mathrm{nnz}(S) + d + N))$.

It remains to show that $(\widetilde{\tau}_i)_{i=1}^N$ and $\widetilde{q}$ as defined in Equations 7 and 8 can be computed quickly. Note that $\mathrm{tr}(AA^\top)$ is the entrywise inner product of $A$ with itself, so it can be computed in time $O(rd)$. The vectors $(A(X_i - \mu(w^t)))_{i=1}^N$ can be computed in time $O\left(r \cdot (\sum_i \mathrm{nnz}(X_i) + d)\right) = O(r \cdot (\mathrm{nnz}(S) + d))$, because each $AX_i$ can be compute in time $O(r \cdot \mathrm{nnz}(X_i))$ and $A\mu(w^t)$ can be computed only once in time $O(rd)$. Because $r = O(\log N \log(1/\delta))$, we can compute all $\widetilde{\tau}_i$ in time $O(r \cdot (\mathrm{nnz}(S) + d))$. Given the $\widetilde{\tau}_i$'s, $\widetilde{q}$ can be computed in $O(N)$ time.

Recall that $r = O(\log N \log(1/\delta))$ and $\ell = O(\log d)$. Putting everything together, the overall runtime of the oracle is

$$O(r \cdot \ell \cdot t \cdot (\mathrm{nnz}(S) + d + N)) + O(r \cdot (\mathrm{nnz}(S) + d) + N) = \widetilde{O}(t \cdot (\mathrm{nnz}(S) + d + N) \cdot \log(1/\delta)) . \quad \square$$

By Lemma C.4 and the fact that $t = O(\log d)$ and $\delta = 1/\mathrm{poly}(d)$, we can implement an approximate score oracle that succeeds with high probability and runs in time $\widetilde{O}(\mathrm{nnz}(S) + N + d)$.

Lemma 3.3 follows from Lemma C.1 and the correctness and the runtime of the approximate score oracle (Lemmas C.3, and C.4). (Note that we have $\mathrm{nnz}(S) = Nd$, $N = N$, and $d = m$ when invoking these lemmas.)

## C.1 Proof of Lemma C.1

In this section, we will show the equivalence between Lemma C.1 and Lemma C.6. Lemma C.6 is a restatement of the result of Dong et al. (2019) using their stability notations.

We first state the stability condition used throughout Dong et al. (2019).

**Definition C.5** (Dong et al. (2019))**.** *We say a set of points $S$ is $(\epsilon, \gamma_1, \gamma_2, \beta_1, \beta_2)$-good with respect to a distribution $D$ with true mean $\mu$ if the following two properties hold:*

- $\|\mu(S) - \mu\|_2 \le \gamma_1$, $\left\|\frac{1}{|S|}\sum_{i \in S}(X_i - \mu(S))(X_i - \mu(S))^\top - I\right\|_2 \le \gamma_2$.

- *For any subset $T \subset S$ so that $|T| = 2\epsilon|S|$, we have*

$$\left\|\frac{1}{|T|}\sum_{i \in T}X_i - \mu\right\|_2 \le \beta_1, \left\|\frac{1}{|T|}\sum_{i \in T}(X_i - \mu(S))(X_i - \mu(S))^\top - I\right\|_2 \le \beta_2.$$

Then, Dong et al. (2019) showed the following result.

**Lemma C.6.** *Let $D$ be a distribution on $\mathbb{R}^d$ with unknown mean $\mu$. Let $0 < \epsilon < \epsilon_0$ for some universal constant $\epsilon_0$. Let $S$ be a set of $N$ samples with $S = S_g \cup S_b \backslash S_r$ where $|S_b|, |S_r| \le \epsilon|S|$, and $S_g$ is $(\epsilon, \gamma_1, \gamma_2, \beta_1, \beta_2)$-good with respect to $D$. Let $\mathscr{O}_{apx}$ be an approximate score oracle for $S$. Suppose $\|X\|_2 \le R$ for all $X \in S$. Then, there is an algorithm QUEScoreFilter(S, $\mathscr{O}_{apx}$, $\delta$) that outputs a $\hat{\mu}$ such that with high probability,*

$$\|\hat{\mu} - \mu\|_2 \le O(\epsilon\sqrt{\log 1/\epsilon} + \sqrt{\epsilon\xi} + \gamma_1),$$

*where*

$$\xi = \gamma_2 + 2\gamma_1^2 + 4\epsilon^2\beta_1^2 + 2\epsilon\beta_2 + O(\epsilon\log 1/\epsilon).$$

*Moreover, QUEScoreFilter makes $O(\log R \log d)$ calls to the score oracle $\mathscr{O}_{apx}$, and the rest of the algorithm runs in time $\widetilde{O}(N\log(R))$.*

We first show the connection between our stability notion (Definition 2.5) and theirs (Definition C.5).

**Lemma C.7.** *Fix a $d$-dimensional distribution $D$ with mean $\mu$, if a set $S$ of $N$ samples is $(\epsilon, \beta, \gamma)$-stable with respect to $D$, then $S$ is $(\epsilon, \beta, \gamma, \beta/\epsilon, \gamma/\epsilon + 3\beta^2/\epsilon^2)$-good with respect to $D$.*

*Proof.* For any subset $T \subset S$ with $|T| = 2\epsilon|S|$, we have

$$\left\|\frac{1}{|T|}\sum_{i \in T}X_i - \mu\right\|_2 = \left\|\left(\frac{1}{|T|}\sum_{i \in S}X_i - \frac{1}{2\epsilon}\mu\right) - \left(\frac{1}{|T|}\sum_{i \in S\backslash T}X_i - \left(\frac{1}{2\epsilon} - 1\right)\mu\right)\right\|_2$$

$$= \left\|\frac{1}{2\epsilon}\left(\frac{1}{N}\sum_{i \in S}X_i - \mu\right) - \frac{(1 - 2\epsilon)}{2\epsilon}\left(\frac{1}{(1 - 2\epsilon)N}\sum_{i \in S\backslash T}X_i - \mu\right)\right\|_2$$

$$\le \frac{1}{2\epsilon}\left\|\frac{1}{N}\sum_{i \in S}X_i - \mu\right\|_2 + \frac{1 - 2\epsilon}{2\epsilon}\left\|\frac{1}{(1 - 2\epsilon)N}\sum_{i \in S\backslash T}X_i - \mu\right\|_2$$

$$\le \frac{1}{2\epsilon}\beta + \frac{1}{2\epsilon}\beta = \frac{\beta}{\epsilon}.$$

The last line follows from the assumption that $S$ is $(\epsilon, \beta, \gamma)$-stable with respect to $D$.

Similarly, we have

$$\left\|\frac{1}{|T|}\sum_{i \in T}(X_i - \mu)(X_i - \mu)^\top - I\right\|_2$$

$$= \left\|\frac{1}{2\epsilon}\left(\frac{1}{N}\sum_{i \in S}(X_i - \mu)(X_i - \mu)^\top - I\right) - \frac{(1 - 2\epsilon)}{2\epsilon}\left(\frac{1}{(1 - 2\epsilon)N}\sum_{i \in S\backslash T}(X_i - \mu)(X_i - \mu)^\top - I\right)\right\|_2$$

$$\le \frac{1}{2\epsilon}\left\|\frac{1}{N}\sum_{i \in S}(X_i - \mu)(X_i - \mu)^\top - I\right\|_2 + \frac{1 - 2\epsilon}{2\epsilon}\left\|\frac{1}{(1 - 2\epsilon)N}\sum_{i \in S\backslash T}(X_i - \mu)(X_i - \mu)^\top - I\right\|_2$$

$$\le \frac{1}{2\epsilon}\gamma + \frac{1}{2\epsilon}\gamma = \frac{\gamma}{\epsilon}.$$

Notice that

$$
\frac{1}{|T|} \sum_{i \in T} (X_i - \mu(S))(X_i - \mu(S))^\top
$$

$$
= \frac{1}{|T|} \sum_{i \in T} (X_i - \mu)(X_i - \mu)^\top + (\mu - \mu(S))(\mu - \mu(S))^\top
$$

$$
+ (\mu - \mu(S)) \left( \frac{1}{|T|} \sum_{i \in T} X_i - \mu \right)^T + \left( \frac{1}{|T|} \sum_{i \in T} X_i - \mu \right) (\mu - \mu(S))^T .
$$

Combining the above two inequalities, by the triangle inequality, we get

$$
\left\| \frac{1}{|T|} \sum_{i \in T} (X_i - \mu(S))(X_i - \mu(S))^\top - I \right\|_2 \leq \frac{\gamma}{\epsilon} + \|\mu - \mu(S)\|_2^2 + \frac{2\beta}{\epsilon} \|\mu - \mu(S)\|_2
$$

$$
= \frac{\gamma}{\epsilon} + \frac{3\beta^2}{\epsilon^2} . \qquad \square
$$

When $S$ is $(\epsilon, \beta, \gamma)$-stable, by Lemma C.6, we know that $S$ is $(\epsilon, \gamma_1 = \beta, \gamma_2 = \gamma, \beta_1 = \beta/\epsilon, \beta_2 = \gamma/\epsilon + 3\beta^2/\epsilon^2)$-good. The parameter $\xi$ in Lemma C.7 is

$$
\xi = 3\gamma + 6\beta^2 + 6\beta^2/\epsilon ,
$$

and error guarantee in Lemma C.7 translates to $\sqrt{\epsilon\xi} + \gamma_1 = O(\sqrt{\epsilon\gamma} + \beta)$, which is exactly what is needed in Lemma C.6.

## D    OMITTED PROOFS FROM SECTION 4

In this section, we prove the technical lemmas in Section 4. We restate each lemma before proving it.

Lemma 4.2 states that the (scaled) initial estimation is not too far from the true conditional probabilities $p$.

**Lemma 4.2.** *Consider the same setting as in Theorem 4.1. Assume the conditions in Section 2.3 hold. In Algorithm 1, we have*

$$
\left\| \pi^P \circ (\hat{p} - \hat{q}^0) \right\|_2 \leq O(\epsilon \sqrt{d} / \sqrt{\alpha c}) .
$$

*Proof.* Recall that $q^0 = q^S$ is the empirical conditional probabilities over $S$, and $\hat{v} = v \circ s$ where $s$ is the scaling vector with $\|s\|_\infty \leq O(1/\sqrt{\alpha c})$.

Let $\pi^{G^\star}$ and $p^{G^\star}$ be the empirical parental configuration probabilities and conditional probabilities given by $G^\star$.

We first show that

$$
\left\| \pi^{G^\star} - \pi^S \right\|_2 \leq \epsilon \sqrt{2d} .
$$

Let $n_k^{G^\star}$ and $n_k^S$ denote the number of times that $\Pi_k$ happens in $G^\star$ and $S$. Note that changing one sample in $G^\star$ can increase or decrease $n_k^{G^\star}$ by at most 1. Moreover, in a single sample, exactly $d$ parental configuration events happen, so changing a sample can affect at most $2d$ $n_k^{G^\star}$'s. Since $S$ is obtained from $G^\star$ by changing $\epsilon N$ samples, we have $|n_k^{G^\star} - n_k^S| \leq \epsilon N$ for all $k$, and $\sum_k |n_k^{G^\star} - n_k^S| \leq 2\epsilon d N$. Together they imply $\left\| \pi^{G^\star} - \pi^S \right\|_2 \leq \epsilon \sqrt{2d}$.

By a similar argument, we can show that

$$
\left\| \pi^{G^\star} \circ p^{G^\star} - \pi^S \circ q^S \right\|_2 \leq \epsilon \sqrt{2d} ,
$$

because $\pi_k^{G^\star} p_k^{G^\star}$ is the probability that $\Pi_k$ happens and $X(k) = 1$ over $G^\star$.

By the triangle inequality, we have

$$\left\|\pi^{G^\star} \circ (p^{G^\star} - q^S)\right\|_2 \leq \left\|\pi^{G^\star} \circ p^{G^\star} - \pi^S \circ q^S\right\|_2 + \left\|\pi^{G^\star} - \pi^S\right\|_2 \leq 3\epsilon\sqrt{d} \ .$$

Using the condition in Equation equation 1 from Section 2.3, i.e., $\left\|\pi^{G^\star} \circ (p^{G^\star} - p)\right\|_2 \leq O(\epsilon)$, we get

$$\left\|\pi^{G^\star} \circ (p - q^S)\right\|_2 \leq O(\epsilon\sqrt{d}) \ .$$

Now by Equation equation 2 from Section 2.3 and the assumption that the minimum parental configuration probability $\min_k \pi_k^P = \alpha = \Omega(\epsilon)$, we have $\pi_k^P \leq \pi_k^{G^\star} + O(\epsilon) \leq O(\pi_k^{G^\star})$, and hence

$$\left\|\pi^P \circ (p - q^S)\right\|_2 \leq O(\epsilon\sqrt{d}) \ .$$

After scaling by $s$, we have

$$\left\|\pi^{G^\star} \circ (\hat{p} - \hat{q}^S)\right\|_2 \leq O(\epsilon\sqrt{d}/\sqrt{\alpha c}) \ . \hspace{2cm} \square$$

Lemma 4.3 shows that, when $q$ is relatively far from $p$, the algorithm can find a new $q$ such that $\left\|\pi^P \circ (\hat{p} - \hat{q})\right\|_2$ decreases by a constant factor.

**Lemma 4.3.** *Consider the same setting as in Theorem 4.1. Assume the conditions in Section 2.3 hold. Fix an iteration $t$ in Algorithm 1. Assume the robust mean estimation algorithm $\mathscr{A}_{mean}$ succeeds. If $\left\|\pi^P \circ (\hat{p} - \hat{q}^t)\right\|_2 \leq \rho^t$ and $\rho^t = \Omega(\epsilon\sqrt{\log(1/\epsilon)}/\sqrt{\alpha c})$, then we have*

$$\left\|\pi^P \circ (\hat{p} - \hat{q}^{t+1})\right\|_2 \leq c_1 \rho^t$$

*for some universal constant $c_1 < 1$.*

*Proof.* We assume $\rho^t > c_4(\epsilon\sqrt{\log(1/\epsilon)}/\sqrt{\alpha c})$ and $\alpha > c_5\epsilon$ for some sufficiently large universal constants $c_4$ and $c_5$.

Because $\left\|\pi^P \circ (\hat{p} - \hat{q}^t)\right\|_2 \leq \rho^t$, Lemma 3.2 shows that $\left\{\hat{f}(X_i, q^t)\right\}_{i \in G^\star}$ is

$$\left(\epsilon, O\left(\frac{\epsilon\sqrt{\log 1/\epsilon}}{\sqrt{\alpha c}} + \frac{\epsilon}{\alpha}\rho^t\right), O\left(\frac{\epsilon\log 1/\epsilon}{\alpha c} + \frac{(\rho^t)^2}{\alpha} + \frac{\rho^t}{\sqrt{\alpha}}\right)\right)\text{-stable.}$$

By Lemma 3.3, the robust mean estimation oracle $\mathscr{A}_{mean}$, which we assume to succeed, outputs a $\nu \in \mathbb{R}^m$ such that, for some universal constant $c_3$,

$$\left\|\nu - \pi^P \circ (\hat{p} - \hat{q}^t)\right\|_2 \leq c_3 \left(\sqrt{\frac{\epsilon}{\alpha}}\rho^t + \sqrt{\frac{\epsilon}{\sqrt{\alpha}}}\rho^t + \frac{\epsilon}{\alpha}\rho^t + \frac{\epsilon\sqrt{\log(1/\epsilon)}}{\sqrt{\alpha c}}\right)$$

$$< \left(\frac{c_3}{\sqrt{c_5}} + \frac{c_3}{\sqrt{c_4}} + \frac{c_3}{c_5} + \frac{c_3}{c_4}\right)\rho^t \ .$$

From Section 2.3, we have $\left\|\pi^S - \pi^P\right\|_\infty = O(\epsilon)$, which implies

$$\left\|(\pi^S - \pi^P) \circ (\hat{p} - \hat{q}^t)\right\|_2 \leq \frac{\epsilon}{\alpha}\left\|\pi^P \circ (\hat{p} - \hat{q}^t)\right\|_2 \leq \frac{\epsilon}{\alpha}\rho^t \ .$$

By the triangle inequality, we have

$$\left\|\nu - \pi^S \circ (\hat{p} - \hat{q}^t)\right\|_2 \leq \left(\frac{c_3}{\sqrt{c_5}} + \frac{c_3}{\sqrt{c_4}} + \frac{c_3 + 1}{c_5} + \frac{c_3}{c_4}\right)\rho^t \ .$$

Algorithm 1 sets $\hat{q}^{t+1} = \nu \circ (1/\pi_S) + \hat{q}^t$, which is equivalent to

$$\pi^S \circ (\hat{p} - \hat{q}^{t+1}) = \pi^S \circ (\hat{p} - \hat{q}^t) - \nu \ .$$

Since $\left\|\pi^S - \pi^P\right\|_\infty = O(\epsilon)$ and $\alpha = \Omega(\epsilon)$, we have

$$\pi_i^P \leq 1.1\pi_i^S \quad \forall 1 \leq i \leq m \ .$$

Putting everything together, letting $c_1 = 1.1\left(\frac{c_3}{\sqrt{c_5}} + \frac{c_3}{\sqrt{c_4}} + \frac{c_3+1}{c_5} + \frac{c_3}{c_4}\right)$, we have

$$\left\|\pi^P \circ (\hat{p} - \hat{q}^{t+1})\right\|_2 \leq 1.1\left\|\pi^S \circ (\hat{p} - \hat{q}^{t+1})\right\|_2 < 1.1\left(\frac{c_3}{\sqrt{c_5}} + \frac{c_3}{\sqrt{c_4}} + \frac{c_3+1}{c_5} + \frac{c_3}{c_4}\right)\rho^t = c_1\rho^t \ .$$

Because $c_4$ and $c_5$ can be sufficiently large, we have $c_1 < 1$ as needed. $\qquad\square$

Lemma 4.4 shows that when the algorithm terminates, we can conclude that the output $Q$ is close to the ground-truth $P$ in total variation distance.

**Lemma 4.4.** *Consider the same setting as in Theorem 4.1. Assume the conditions in Section 2.3 hold. Let $Q$ be a Bayesian network that shares the same structure with $P$. Suppose that (1) $P$ is c-balanced, (2) $\alpha = \Omega(r + \epsilon/c)$, and (3) $\left\|\pi^P \circ (\hat{p} - \hat{q})\right\|_2 \leq r/2$. Then we have*

$$d_{\mathrm{TV}}(P, Q) \leq r \ .$$

*Proof of Lemma 4.4.* We have $(p_k + q_k)(2 - p_k - q_k) \geq p_k(1 - p_k)$. Hence,

$$\sum_k \sqrt{\pi_k^P \pi_k^Q}\frac{(p_k - q_k)^2}{(p_k + q_k)(2 - p_k - q_k)} \leq \sum_k \sqrt{\pi_k^P \pi_k^Q}\pi_k^P\frac{(p_k - q_k)^2}{\pi_k^P p_k(1 - p_k)} \ ,$$

From the proof of Lemma B.4, we know $|\pi_k^P p_k(1-p_k) - \frac{1}{s_k^2}| = O(\epsilon)$ and $\pi_k^P p_k(1-p_k) \geq \pi_k^P\frac{p_k}{2} = \Omega(r)$, so we have

$$\sum_k \sqrt{\pi_k^P \pi_k^Q}\pi_k^P\frac{(p_k - q_k)^2}{\pi_k^P p_k(1 - p_k)} \leq 1.1\sum_k \sqrt{\pi_k^P \pi_k^Q}\pi_k^P(p_k - q_k)^2 s_k^2$$

$$= 1.1\sum_k \sqrt{\pi_k^P \pi_k^Q}\pi_k^P(\hat{p}_k - \hat{q}_k)^2 \ .$$

It suffices to show that $|\pi_k^P - \pi_k^Q| \leq r$, which implies $\pi_k^Q \leq 1.1\pi_k^P$ and further implies

$$d_{\mathrm{TV}}(P, Q) \leq 2\left\|\pi^P \circ (\hat{p} - \hat{q})\right\|_2 \ .$$

Let $P_{\leq i}$ and $Q_{\leq i}$ be the distributions of the first $i$ coordinates of $P$ and $Q$ respectively. We prove $|\pi_k^P - \pi_k^Q| \leq r$ by induction on $i$. Suppose that for $1 \leq j < i$ and all $a' \in \{0, 1\}^{|\mathrm{parents}(j)|}$, $|\pi_{j,a'}^P - \pi_{j,a'}^Q| \leq r$, then we have $d_{\mathrm{TV}}(P_{\leq(i-1)}, Q_{\leq(i-1)}) \leq r$. Because that events $\Pi_{i,a}$ only depends on $j < i$, $|\pi_{i,a}^P - \pi_{i,a}^Q| \leq d_{\mathrm{TV}}(P_{\leq(i-1)}, Q_{\leq(i-1)}) \leq r$ for all $a$. Consequently, we have $d_{\mathrm{TV}}(P, Q) = d_{\mathrm{TV}}(P_{\leq d}, Q_{\leq d}) \leq r$. $\qquad\square$

