# OpenReview forum: "Robust Learning of Fixed-Structure Bayesian Networks in Nearly-Linear Time"
_ICLR.cc/2021/Conference — ICLR 2021 Poster_

### Official Review · AnonReviewer4 · 2020-10-26
**Initial review**

**Rating:** 7
**Confidence:** 3

**Review:**

**Summary**
This work shows how to learn a ground truth Bayesian network where an
$\epsilon$-fraction of the samples are adversarially corrupted. The authors
focus on the fully observable case when all the variables are binary and the
underlying graph is given to the algorithm. The main results in this paper are
(1) a nearly-linear time algorithm for this problem with a dimension-independent
error guarantee, and (2) a direct connection between robust mean estimation and
learning Bayesian networks. The authors achieve this by extending the robust
mean estimation algorithm of [Dong et al., NeurIPS 2019] to handle sparse input.

**Strengths**
This paper is technically very strong. Learning Bayesian networks is becoming
an increasingly important problem, as is the need for robust algorithms. One of
the main contributions in this work is the direct connection to robust mean
estimation, which has received a flurry of attention in the past several years.
Minimizing the spectral norm of the covariance matrix (as opposed to trying to
make it diagonal) is also a nice deviation from [Cheng et al., NeurIPS 2018].
The well-planned presentation of the results is an added bonus.

**Weaknesses**
While this is primarily a theory paper, a set of accompanying experiments would
be nice. Applications of learning Bayesian networks could also be discussed in
more detail in the introduction of the paper.

**Suggestions**
- [ 1] Typo: "corruption" --> "corruptions"
- [ 2] Typo: Extra parenthesis at the end of Theorem 1.2.
- [ 2] Typo: "previous work" --> "previous works"
- [ 2] Typo: "Previous algorithm" --> "Previous algorithms"
- [ 2] Typo: "either has" --> "either have"
- [ 3] It might be helpful to use 'n' instead of 'N' for the number of samples.
- [ 3] Typo: X \in \R^{N \times d} should be X \in \R^{d \times N}.

---

> ### Author Response · Authors · 2020-11-19
> **We thank the reviewer for the encouraging and helpful comments.**
>
> We thank the reviewer for the encouraging and helpful comments.  We will correct the typos mentioned.
>
> We agree with the reviewer that it would be nice to include some experiments.  One of the reasons we did not include any experiments is that current nearly-linear time robust mean estimation algorithms require subroutines like matrix multiplicative weight update and the JL lemma which may not be very efficient in practice.  Due to this reason, the experiments in [Dong-Hopkins-Li] mainly focused on the effectiveness of QUE-scoring vs. other outlier detection methods, rather than a runtime comparison.

---

### Official Review · AnonReviewer1 · 2020-11-01
**A bit incremental and derivative from prior work**

**Rating:** 5
**Confidence:** 4

**Review:**

The paper studies the problem of robust learning of fixed-structure Bayesian networks under the eps-adversarial corruptions model. Fixed-structure means a known structure of the underlying Bayesian network. Robust learning is an important area of research and this particular question has been studied in prior work. The main contribution of this work is in improving the running time of the algorithm. On a d-node Bayes net, let m denote the total number of parental configurations possible. Prior work of Cheng et al showed a robust learning algorithm using O(m/eps^2) samples and runs in time O(md^2/eps^2).
The current paper reduces the running time to O(md/eps^2). The core subroutine is a robust mean finding algorithm for general distributions. A nearly linear time algorithm for this problem was recently given in the work of Dong et al which the current paper uses.
The ideas in the current paper use similar constructions to those of Cheng et al and seem to rely on the improvement in the running time of robust mean estimation provided by the work of Dong et al. Given this, I feel, although the paper has a clear improvement on a result for a specific problem it is a bit derivative.

Minor typos:
1. Lemma 2.6: “these” -> “there”
2. Page 6: intuition for Lemma B.3 (should be Lemma 3.2?)



---------------
I thank the authors for their response. However, I am inclined to keep my rating after having read their response.

---

> ### Author Response · Authors · 2020-11-19
> **We thank the reviewer for the helpful comments.**
>
> We thank the reviewer for the helpful comments.  We will correct the typos mentioned.
>
> One of our main contributions is to derive a more explicit connection between robust learning of Bayes nets and robust mean estimation.  This connection was not explicit at all in previous works.  For example, [Cheng-Diakonikolas-Kane-Stewart] used a filtering-based approach that is more tailored to binary product distributions and proved specific tail bounds on the modified samples.  In contrast, our result allows one to take advantage of any stability-based robust mean estimation algorithm (e.g., new algorithms that are simpler/faster or work better in practice) and apply it almost directly to solve robust learning of Bayes nets.
>
> The previous algorithm by Cheng et al runs in time \tildeO(m d^2 / eps^3), not eps^2.

---

### Official Review · AnonReviewer5 · 2020-11-08
**Paper leverages connections  to robust mean estimation for robustly learning Bayesian networks in near-linear time, but the current contribution appears to be a bit limited.**

**Rating:** 4
**Confidence:** 4

**Review:**

The paper considers the problem of robustly learning fixed structure Bayesian networks in nearly-linear time. Previous work by Cheng et al. gives a runtime of O(Nd^2/eps). The paper improves this to O(Nd). The algorithm works by directly relating the problem to robust mean estimation, and then leveraging the algorithm of Dong et al. for robust mean estimation which works in nearly-linear time. The authors have to modify the runtime analysis of the algorithm of Dong et al. to work in time linear in the sparsity, rather than dimension.

I found the current contribution to be a bit limited. The connection between robust mean estimation and robustly learning Bayesian networks is already made and leveraged in Cheng et al. The new contribution here is to show that robust mean estimation can be used as a black-box. This is done by showing that the recent stability condition established to be sufficient for robust mean estimation can be satisfied for robustly learning Bayesian networks by scaling the moments by empirical estimates, for example scaling the covariance to be identity.

Once the reduction is established, the the paper leverages the algorithm of Dong et al. for robust mean estimation. However, the vectors arising in learning Bayesian networks are sparse, but Dong et al. does not exploit this sparsity. Dong et al. does run in time linear in sparsity, except for calls to a score oracle. The scores are computed in Dong et al. using the JL lemma. This papers shows that this part can also be done in time linear in sparsity. A lot of work has shown that JL style dimensionality reduction can be done in  input sparsity time, I don't see why these cannot be directly applied here (for example Clarkson-Woodruff, "Low Rank Approximation and Regression in Input Sparsity Time")?

I think if the authors can extend the results and the proposed black-box reduction/input sparsity time robust mean estimation to apply to other regimes of robustly learning Bayesian networks, or some other problems, then it would be interesting. I also think that it would be good if the paper had experiments to back the algorithm---it would especially make the paper more interesting and relevant to the ICLR audience. Note that the algorithms of both Cheng et al. and Dong et al. are practical and the papers have experimental evaluations. If the authors can show that their algorithm empirically improves on the performance of Cheng et al. and other approaches that they compare to, then that would make the paper stronger.

----------Update after author response----------

I thank the authors for the detailed response. I think the fact that the near-linear time JL approach follows more or less from previous work needs to be clearly mentioned in the paper. I also think some experiments would be nice, and it would be reasonable to use some of the heuristics which the authors suggested, and sparse JL can often be reasonably efficient in practice. In light of all this I am keeping my score, but would encourage the authors to perhaps further pursue these directions.

---

> ### Author Response · Authors · 2020-11-19
> **We thank the reviewer for the careful consideration of our paper.**
>
> We thank the reviewer for the careful consideration of our paper.
>
> One of our main contributions is to derive a more explicit connection between robust learning of Bayes nets and robust mean estimation.  This connection was not explicit at all in previous works.  For example, [Cheng-Diakonikolas-Kane-Stewart] used a filtering-based approach that is more tailored to binary product distributions and proved specific tail bounds on the modified samples.  In contrast, our result allows one to take advantage of any stability-based robust mean estimation algorithm (e.g., new algorithms that are simpler/faster or work better in practice) and apply it almost directly to solve robust learning of Bayes nets.
>
> We agree with the reviewer that there are examples of input-sparsity time Johnson-Lindenstrauss (JL) in many related works.  However, previous work did not prove a result of the form "if JL can be done in input-sparsity time, then robust mean estimation can be done in nearly-linear time".  We state this result explicitly (our Lemma 3.3).  While such a result may not be surprising given [Dong-Hopkins-Li], and one could even consider stating it without giving a formal proof (using the argument provided by the reviewer), we decided to be self-contained and filled in all the technical details.
>
> We agree with the reviewer that it would be nice to include some experiments.  One of the reasons we did not include any experiments is that current nearly-linear time robust mean estimation algorithms require subroutines like matrix multiplicative weight update and the JL lemma which may not be very efficient in practice.  Due to this reason, the experiments in [Dong-Hopkins-Li] mainly focused on the effectiveness of QUE-scoring vs. other outlier detection methods, rather than a runtime comparison;  In fact, in some of the larger-scale experiments, [Dong-Hopkins-Li] proposed heuristics for approximating QUE-scores without JL (see Section 7.2 of their arXiv version).

---

### Official Review · AnonReviewer2 · 2020-11-09

**Rating:** 7
**Confidence:** 4

**Review:**

OVERVIEW
==============

This paper studies the problem of learning Bayes nets using adversarially corrupted data. The model is that $N$ samples are made from a Bayes net on d nodes, out of which an unknown $\varepsilon$ fraction are changed arbitrarily. The structure of the Bayes net is already given, but it remains to learn the probability distribution.

When there are no corruptions, it is well known that in $O(N)$ time (for sufficiently large $N$), one can estimate the Bayes net upto TV distance $\varepsilon$. This is without any extra assumptions.

In this paper, it's shown that in the above corruption model, there is an algorithm requiring $\tilde{O}(Nd)$ time that learns the Bayes net within TV distance $\varepsilon \cdot \sqrt{\ln(1/\varepsilon)}$. However, they also require two extra assumptions of "balanced"-ness and "minimum parental configuration probability". The previous best running time (under the same assumptions) was $\tilde{O}(Nd^2)$.

The key contribution of the work is that they show a clean reduction to robust mean estimation. Unlike previous work on this problem, they make their algorithm super simple and natural, at the expense of making their analysis somewhat more complicated.

Technically, there are two new innovations compared to the earlier work. The first is a data-adaptive scaling which allows them to prove a certain stability property of an expanded conditional probability table. The second is a speed-up for robust mean estimation that runs in time nearly linear in the number of nonzeros in the input.

EVALUATION
============

(There is a technical point which seems to be not discussed. In Lemma 3.2, what if s blows up to infinity because q^S at some particular (i,a) is almost 0 or 1? I don't believe this is a fatal issue but it should be clarified.)

Positives:
* solves a natural high-dimensional inference problem in the robust setting
* improvement to the running time uses nice ideas that may have other applications
* paper is well-written

Negatives:
* Given that this is ICLR, I'd have liked to see some experiments validating the improvement in runtime.
* The assumptions are taken completely for granted in this paper, while they are very artificial and clearly not needed in the non-robust setting. I'd have liked more discussion as to why they are necessary for the analysis (perhaps, a counter-example which makes a step in their analysis fail?)

Overall, I am in favor of accepting the paper.

MORE DETAILED COMMENTS
========================

* Page 1, second para of intro: I'd change "fundamental" to "simplest". One could argue structure learning is more 'fundamental'.

* Page 1, next-to-last para: Actually, Dasgupta doesn't exactly use the empirical estimator. He "shifts" the empirical distribution so that probabilities are bounded away from 0 and 1.

* Page 4, second para: Balancedness is in fact not necessary for closeness in conditional probabilities to imply closeness in the global distribution (this is why Dasgupta's analysis works)

* Page 6, second para: S is not defined here. Mention that it's the set of corrupted samples.

* Page 7, Lemma 4.3 and elsewhere: Please change the > O(.) notation to > \Theta(.).

---

> ### Author Response · Authors · 2020-11-19
> **We thank the reviewer for the detailed and encouraging comments.**
>
> We thank the reviewer for the detailed and encouraging comments.  We will correct the typos mentioned.
>
> Regarding the technical question on the vector s in Definition 3.1:  At a high level, s would not blow up because we assumed "balanced"-ness and "minimum parental configuration probability".  More specifically, the empirical value of (pi^S \circ q^S \circ (1-q^S)) must be eps-close to the true value (pi^P \circ p \circ (1-p)) (see, e.g., Lemma B.4).  The later is at least (alpha * c / 2), so for example when alpha and c are both \Omega(sqrt(eps)), the empirical estimate will be within (1 +/- 0.1) of the true value.  This is one of the places why we require the minimum parental configuration probability alpha to be larger than some function of c and eps (see the requirement in Theorem 4.1).  We will clarify this.

---

### Decision · Program_Chairs · 2021-01-07
**Final Decision**

**Decision:**

Accept (Poster)

**Comment:**

The main contribution of this paper is a nearly linear time algorithm for learning Bayesian networks with a known structure when an epsilon fraction of the samples are contaminated. The model assumes that the directed graph is known and the goal is to estimate a vector of length m that describes the conditional distribution of any node for any configuration if its parents. Let N be the number of samples and let d be the number of nodes. Prior work gave an algorithm that runs in time N d^2 time. This is now improved to roughly Nd time under natural conditions on the "balancedness" and the "minimum parental configuration probability". The algorithm itself is simple, and is a more direct reduction to robust mean estimation.

The reviewers had somewhat differing opinions. The pros are that it's a basic problem, the algorithm is clean and the ingredients in the improved running time could have further applications. The negative is that there are no experiments, even synthetic ones, to demonstrate practicality. Overall it still seems that there is enough excitement about the work to merit acceptance.